# Optimal Excess Risk Bounds for Empirical Risk Minimization on $p$-Norm Linear Regression

**Ayoub El Hanchi**
University of Toronto & Vector Institute
`aelhan@cs.toronto.edu`

**Murat A. Erdogdu**
University of Toronto & Vector Institute
`erdogdu@cs.toronto.edu`

## Abstract

We study the performance of empirical risk minimization on the $p$-norm linear regression problem for $p \in (1, \infty)$. We show that, in the realizable case, under no moment assumptions, and up to a distribution-dependent constant, $O(d)$ samples are enough to exactly recover the target. Otherwise, for $p \in [2, \infty)$, and under weak moment assumptions on the target and the covariates, we prove a high probability excess risk bound on the empirical risk minimizer whose leading term matches, up to a constant that depends only on $p$, the asymptotically exact rate. We extend this result to the case $p \in (1, 2)$ under mild assumptions that guarantee the existence of the Hessian of the risk at its minimizer.

## 1 Introduction

Real-valued linear prediction is a fundamental problem in machine learning. Traditionally, the square loss has been the default choice for this problem. The performance of empirical risk minimization (ERM) on linear regression under the square loss, as measured by the excess risk, has been studied extensively both from an asymptotic [Whi82; LC83; Vaa98] and a non-asymptotic point of view [AC11; HKZ12; Oli16; LM16; Sau18; Mou22]. An achievement of the last decade has been the development of non-asymptotic excess risk bounds for ERM on this problem under weak assumptions, and which match, up to constant factors, the asymptotically exact rate.

In this paper, we consider the more general family of $p$-th power losses $t \mapsto |t|^p$ for a user-chosen $p \in (1, \infty)$. Under mild assumptions, the classical asymptotic theory can still be applied to ERM under these losses, yielding the asymptotic distribution of the excess risk. However, to the best of our knowledge, the problem of deriving non-asymptotic excess risk bounds for ERM for $p \in (1, \infty) \setminus \{2\}$ remains open, and, as we discuss below, resists the application of standard tools from the literature.

Our motivation for extending the case $p = 2$ to $p \in (1, \infty)$ is twofold. Firstly, the freedom in the choice of $p$ allows us to better capture our prediction goals. For example, we might only care about how accurate our prediction is on average, in which case, the choice $p = 1$ is appropriate. At the other extreme, we might insist that we do as well as possible on a subset of inputs of probability 1, in which case the choice $p = \infty$ is best. A choice of $p \in (1, \infty)$ therefore allows us to interpolate between these two extremes, with the case $p = 2$ offering a balanced choice. Secondly, different choices of $p$ have complementary qualities. On the one hand, small values of $p$ allow us to operate with weak moment assumptions, making them applicable in more general cases. On the other, larger values of $p$ yield predictions whose optimality is less sensitive to changes in the underlying distribution: for $p = \infty$, the best predictor depends only on the support of this distribution.

To sharpen our discussion, let us briefly formalize our problem. There is an input random vector $X \in \mathbb{R}^d$ and output random variable $Y \in \mathbb{R}$, and we are provided with $n$ i.i.d. samples $(X_i, Y_i)_{i=1}^n$. We select our set of predictors to be the class of linear functions $\{x \mapsto \langle w, x \rangle \mid w \in \mathbb{R}^d\}$, and choose a value $p \in (1, \infty)$ with the corresponding loss $\ell_p(t) := |t|^p / [p(p-1)]$. Using this loss, we define

37th Conference on Neural Information Processing Systems (NeurIPS 2023).

the associated risk and empirical risk, respectively, by

$$R_p(w) := \mathrm{E}[\ell_p(\langle w, X \rangle - Y)], \qquad R_{p,n}(w) := \frac{1}{n}\sum_{i=1}^{n}\ell_p(\langle w, X_i \rangle - Y_i).$$

We perform empirical risk minimization $\hat{w}_p \in \mathrm{argmin}_{w \in \mathbb{R}^d} R_{p,n}(w)$, and our goal is to derive high probability bounds on the excess risk $R_p(\hat{w}_p) - R_p(w_p^*)$, where $w_p^*$ is the risk minimizer. For efficient algorithms for computing an empirical risk minimizer $\hat{w}_p$, we refer the reader to the rich recent literature dealing with this problem [BCLL18; AKPS19; APS19; JLS22].

To see why the problem we are considering is difficult, let us briefly review some of the recent literature. Most closely related to our problem are the results of [AC11; HKZ12; Oli16; LM16], who derive high probability non-asymptotic excess risk bounds for the case $p = 2$. The best such bounds are found in Oliveira [Oli16] and Lecué and Mendelson [LM16], who both operate under weak assumptions on $(X, Y)$, requiring at most the existence of fourth moments of $Y$ and the components $X^j$ of $X$ for $j \in [d]$. Unfortunately, the analysis in Oliveira [Oli16] relies on the closed form expression of the empirical risk minimizer $\hat{w}_2$, and therefore cannot be extended to other values of $p$. Similarly, the analysis in Lecué and Mendelson [LM16] relies on an exact decomposition of the excess loss $\ell_2(\langle w, X \rangle - Y) - \ell_2(\langle w_p^*, X \rangle - Y)$ in terms of "quadratic" and "multiplier" components, which also does not extend to other values of $p$.

To address these limitations, the work of Mendelson [Men18] extends the ideas of Mendelson [Men14] and Lecué and Mendelson [LM16] to work for loss functions more general than the square loss. Roughly speaking, the main result of Mendelson [Men18] states that as long as the loss is strongly convex and smooth in a neighbourhood of $0$, the techniques developed by Mendelson [Men14] can still be applied to obtain high probability excess risk bounds. Unfortunately, the loss functions $\ell_p(t)$ are particularly ill-behaved in precisely this sense, as $\ell_p''(t) \to 0$ when $t \to 0$ for $p > 2$, and $|\ell_p''(t)| \to \infty$ as $t \to 0$ for $p \in (1, 2)$. This makes the analysis of the excess risk of ERM in the case $p \in (1, \infty) \setminus \{2\}$ particularly challenging using well-established methods.

Contrary to the non-asymptotic regime, the asymptotic properties of the excess risk of ERM under the losses $\ell_p(t)$ are better understood [Ron84; BRW92; Nie92; Arc96; HS96; LL05], and can be derived from the more general classical asymptotic theory of $M$-estimators [LC83; VW96; Vaa98] under mild regularity conditions. In particular, these asymptotic results imply that the excess risk of ERM with $n$ samples satisfies

$$\mathrm{E}[R_p(\hat{w}_p)] - R_p(w_p^*) = \frac{\mathrm{E}\Big[\|\nabla \ell_p(\langle w_p^*, X \rangle - Y)\|_{H_p^{-1}}^2\Big]}{2n} + o\bigg(\frac{1}{n}\bigg) \quad \text{as} \quad n \to \infty, \qquad (1)$$

where $H_p := \nabla^2 R_p(w_p^*)$ is the Hessian of the risk at its minimizer. We refer the reader to the discussions in Ostrovskii and Bach [OB21] and Mourtada and Gaïffas [MG22] for more details. As we demonstrate in Theorem 1, the rate of convergence of ERM for the square loss derived in Oliveira [Oli16] and Lecué and Mendelson [LM16] matches the asymptotic rate (1) up to a constant factor. Ideally, we would like our excess risk bounds for the cases $p \in (1, \infty) \setminus \{2\}$ to also match the asymptotic rate (1), although it is not yet clear how to derive any meaningful such bounds.

In this paper, we prove the first high probability non-asymptotic excess risk bounds for ERM under the $p$-th power losses $\ell_p(t)$ for any $p \in (1, \infty) \setminus \{2\}$. Our assumptions on $(X, Y)$ are weak, arise naturally from the analysis, and reduce to the standard ones for the case $p = 2$. Furthermore, the rate we derive matches, up to a constant that depends only on $p$, the asymptotically exact rate (1).

We split the analysis in three cases. The first is when the problem is realizable, i.e. $Y = \langle w^*, X \rangle$ for some $w^* \in \mathbb{R}^d$. This edge case is not problematic for the analysis of the case $p = 2$, but as discussed above, the $\ell_p(t)$ losses are ill-behaved around $0$ for $p \in (1, \infty) \setminus \{2\}$, requiring us to treat this case separately. The second case is when the problem is not realizable and $p \in (2, \infty)$. The final case is when the problem is not realizable and $p \in (1, 2)$, which turns out to be the most technically challenging. In Section 2, we present our main results and in Section 3, we provide their proofs.

**Notation.** We denote the components of the random vector $X \in \mathbb{R}^d$ by $X^j$ for $j \in [d]$. We assume the support of $X$ is not contained in any hyperplane, i.e. $\mathrm{P}(\langle w, X \rangle = 0) = 1$ only if $w = 0$. This is without loss of generality as discussed in Oliveira [Oli16] and Mourtada [Mou22]. For a positive semi-definite matrix $A$, we denote the bilinear form it induces on $\mathbb{R}^d$ by $\langle \cdot, \cdot \rangle_A$, and define $\|\cdot\|_A := \sqrt{\langle \cdot, \cdot \rangle_A}$. We define $H_{p,n} := \nabla^2 R_{p,n}(w_p^*)$.

## 2 Main results

In this section, we state our main results. We start in Section 2.1 where we introduce constants that help us formulate our theorems. In Section 2.2, we state the best known results for both the case $p = 2$ and the realizable case where $Y = \langle w^*, X \rangle$. Finally, in Section 2.3, we state our theorems.

### 2.1 Norm equivalence and small ball constants

To state our results, we will need to define two types of quantities first. The first kind are related to norms and their equivalence constants, which we will use in the analysis of the non-realizable case. The second are small ball probabilities, which we will use in the analysis of the realizable case.

We start by introducing the following functions on our space of coefficients $\mathbb{R}^d$. For $p, q \in [1, \infty)$, define, with the convention $\infty^{1/p} := \infty$ for all $p \in [1, \infty)$,

$$\|w\|_{L^p} := \mathrm{E}[|\langle w, X \rangle|^p]^{1/p}, \qquad \|w\|_{L^q, p} := \mathrm{E}[\|w\|_{\nabla^2 \ell_p(\langle w_p^*, X \rangle - Y)}^q]^{1/q}. \tag{2}$$

As suggested by the notation, under appropriate moment assumptions on $X$, these functions are indeed norms on $\mathbb{R}^d$. In that case, we will be interested in norm equivalence constants between them

$$C_{a \to b} := \sup_{w \in \mathbb{R}^d \setminus \{0\}} \frac{\|w\|_a}{\|w\|_b}, \qquad \sigma_p^2 := C_{(L^4, p) \to (L^2, p)}^4, \tag{3}$$

where $a$ and $b$ stand for one of $L^p$ or $(L^q, p)$. Let us note that since we work in a finite dimensional vector space, all norms are equivalent, so that as soon as the quantities defined in (2) are indeed norms, the constants defined in (3) are finite. Furthermore, as suggested by the notation, $\sigma_p^2$ may be viewed as the maximum second moment of the random variables $\|w\|_{\nabla^2 \ell_p(\langle w_p^*, X \rangle - Y)}^2$ over the unit sphere of $\|\cdot\|_{L^2, p}$. Finally, we record the following identities for future use

$$\|w\|_{L^2, p} = \|w\|_{H_p}, \qquad \|w\|_{L^q, 2} = \|w\|_{L^q}, \qquad \sigma_2^2 = C_{L^4, L^2}^4. \tag{4}$$

The first identity holds by linearity, and the second by noticing that $\nabla^2 \ell_2(\langle w, X \rangle - Y) = XX^T$.

We now turn to small ball probabilities. We define the following functions on $\mathbb{R}^d$, for $q \in [1, \infty)$,

$$\rho_0(w) := \mathrm{P}(\langle w, X \rangle = 0), \qquad \rho_q(w, \kappa) := \mathrm{P}(|\langle w, X \rangle| > \kappa \|w\|_{L^q}). \tag{5}$$

Assumptions on the functions $\rho_0$ and $\rho_2$ have been used extensively in the recent literature, see e.g. [Men14; KM15; LM17a; LM17b; Men18; LM18; Mou22]. In particular, a standard assumption postulates the existence of strictly positive constants $\beta_0$, and $(\beta_2, \kappa_2)$ such that $\rho_0(w) \leq 1 - \beta_0$ and $\rho_2(w, \kappa_2) \geq \beta_2$ for all $w \in \mathbb{R}^d$. Conditions of this type are usually referred to as small ball conditions. Efforts have been made to understand when these conditions hold [Men14; RV15; LM17b] as well as reveal the dimension dependence of the constants with which they do [Sau18]. Here we prove that such conditions always hold for finite dimensional spaces. We leave the proof of Lemma 1 to Appendix B to not distract from our main development.

**Lemma 1.** $\rho_0$ is upper semi-continuous. Furthermore, if for some $q \in [1, \infty)$, $\mathrm{E}[|X^j|^q] < \infty$ for all $j \in [d]$, then $\rho_q(\cdot, \kappa)$ is lower semi-continuous for any $\kappa \geq 0$. Moreover, for all $\kappa \in [0, 1)$

$$\rho := \sup_{w \in \mathbb{R}^d \setminus \{0\}} \rho_0(w) < 1, \qquad \inf_{w \in \mathbb{R}^d \setminus \{0\}} \rho_q(w, \kappa) > 0.$$

### 2.2 Background

To better contextualize our results, we start by stating the best known high probability bound on ERM for the square loss, which we deduce from Oliveira [Oli16] and Lecué and Mendelson [LM16].

**Theorem 1** (Theorem 4.2, Oliveira [Oli16]; Theorem 1.3, Lecué and Mendelson [LM16]). *Assume that* $\mathrm{E}[Y^2] < \infty$ *and* $\mathrm{E}[(X^j)^4] < \infty$ *for all* $j \in [d]$, *and let* $\delta \in (0, 1]$. *If*

$$n \geq 196\sigma_2^2(d + 2\log(4/\delta)),$$

*then, with probability at least* $1 - \delta$

$$R_2(\hat{w}_2) - R_2(w_2^*) \leq \frac{16 \, \mathrm{E}[\|\nabla \ell_2(\langle w_2^*, X \rangle - Y)\|_{H_2^{-1}}^2]}{n\delta}.$$

Up to a constant factor and the dependence on $\delta$, Theorem 1 recovers the asymptotically exact rate (1). Let us briefly comment on the differences between Theorem 1 and the comparable statements in the original papers. First, the finiteness of $\sigma_2^2$ is deduced from the finiteness of the fourth moments of the components of $X$, instead of being assumed as in Oliveira [Oli16] (see the discussion in Section 3.1 in Oliveira [Oli16]). Second we combine Theorem 3.1 from [Oli16] with the proof technique of Lecué and Mendelson [LM16] to achieve a slightly better bound that the one achieved by the proof technique used in the proof of Theorem 4.2 in Oliveira [Oli16], while avoiding the dependence on the small ball-constant present in the bound of Theorem 1.3 in Lecué and Mendelson [LM16], which is known to incur additional dimension dependence in some cases [Sau18].

We now move to the realizable case, where $Y = \langle w^*, X \rangle$ so that $w_p^* = w^*$ for all $p \in (1, \infty)$. We immediately note that Theorem 1 is still applicable in this case, and ensures that we recover $w^*$ exactly with no more than $n = O(\sigma_2^2 d)$ samples. However, we can do much better, while getting rid of all the moment assumptions in Theorem 1. Indeed, it is not hard to see that $\hat{w}_p \neq w^*$ only if for some $w \in \mathbb{R}^d \setminus \{0\}$, $\langle w, X_i \rangle = 0$ for all $i \in [n]$ (taking $w = \hat{w}_p - w_p^*$ works). The implicit argument in Theorem 1 then uses the pointwise bound (see Lemma B.2 in Oliveira [Oli16])

$$\mathrm{P}(\cap_{i=1}^n \{\langle w, X_i \rangle = 0\}) \leq \exp\left(-\frac{n}{2\sigma_2^2}\right),$$

and uniformizes it over the $L^2$ unit sphere in $\mathbb{R}^d$, where the $L^2$ norm is as defined in (2). However, we can use the much tighter bound $\rho^n$ where $\rho$ is as defined in Lemma 1. To the best of our knowledge, the realizable case has not been studied explicitly before in the literature. However, with the above considerations in mind, we can deduce the following result from Lecué and Mendelson [LM17b], which uniformizes the pointwise bound we just discussed using a VC dimension argument.

**Theorem 2** (Corollary 2.5, Lecué and Mendelson [LM17b]). *Assume that there exists $w^* \in \mathbb{R}^d$ such that $Y = \langle w^*, X \rangle$. Let $\delta \in (0, 1]$. If*

$$n \geq O\left(\frac{d + \log(1/\delta)}{(1-\rho)^2}\right)$$

*then for any $p \in (1, \infty)$, $\hat{w}_p = w^*$ with probability at least $1 - \delta$.*

## 2.3 Results

We are now in position to state our main results. As discussed in Section 1, the $\ell_p(t)$ losses have degenerate second derivatives as $t \to 0$. When the problem is realizable, the risk is not twice differentiable at its minimizer for the cases $p \in (1, 2)$, and is degenerate for the cases $p \in (2, \infty)$. If we want bounds of the form (1), we must exclude this case from our analysis. Our first main result is a strengthening of Theorem 2, and relies on a combinatorial argument to uniformize the pointwise estimate discussed in Section 2.2.

**Theorem 3.** *Assume that there exists $w^* \in \mathbb{R}^d$ such that $\langle w^*, X \rangle = Y$. Then for all $n \geq d$, and for all $p \in (1, \infty)$, we have*

$$\mathrm{P}(\hat{w}_p \neq w^*) \leq \binom{n}{d-1} \rho^{n-d+1},$$

*where $\rho$ is as defined in Lemma 1. Furthermore, if*

$$n \geq \begin{cases} O\left(d + \log(1/\delta)/\log(1/\rho)\right) & \text{if} \quad 0 \leq \rho < e^{-1}, \\ O\left(\dfrac{d + \log(1/\delta)}{1-\rho}\right) & \text{if} \quad e^{-1} \leq \rho < e^{-1/e}, \\ O\left(\dfrac{d\log(1/(1-\rho)) + \log(1/\delta)}{1-\rho}\right) & \text{if} \quad e^{-1/e} \leq \rho < 1, \end{cases}$$

*then with probability at least $1 - \delta$, $\hat{w}_p = w^*$.*

Comparing Theorem 2 and Theorem 3, we see that the bound on the number of samples required to reach a confidence level $\delta$ in Theorem 3 is uniformly smaller than the one in Theorem 2. The proof of Theorem 3 can be found in Appendix C.

We now move to the more common non-realizable case. Our first theorem here gives a non-asymptotic bound for the excess risk of ERM under a $p$-th power loss for $p \in (2, \infty)$. To the best of our knowledge, no such result is known in the literature.

**Theorem 4.** *Let $p \in (2, \infty)$ and $\delta \in (0, 1]$. Assume that no $w \in \mathbb{R}^d$ satisfies $Y = \langle w, X \rangle$. Further, assume that $\mathrm{E}[|Y|^p] < \infty$, $\mathrm{E}[|X^j|^p] < \infty$, and $\mathrm{E}[|\langle w_p^*, X \rangle - Y|^{2(p-2)}(X^j)^4] < \infty$ for all $j \in [d]$. If*

$$n \geq 196\sigma_p^2(d + 2\log(4/\delta)),$$

*then with probability at least $1 - \delta$*

$$R_p(\hat{w}_p) - R_p(w_p^*) \leq \frac{2048 p^2 \, \mathrm{E}\left[\|\nabla \ell_p(\langle w_p^*, X \rangle - Y)\|_{H_p^{-1}}^2\right]}{n\delta}$$
$$+ \left(\frac{512 p^4 c_p^2 \, \mathrm{E}\left[\|\nabla \ell_p(\langle w_p^*, X \rangle - Y)\|_{H_p^{-1}}^2\right]}{n\delta}\right)^{p/2},$$

*where we used $c_p$ to denote $C_{L^p \to (L^2, p)}$ as defined in (3).*

Up to a constant factor that depends only on $p$ and the dependence on $\delta$, the bound of Theorem 4 is precisely of the form of the optimal bound (1). Indeed, as $p > 2$, the second term is $o(1/n)$. At the level of assumptions, the finiteness of the $p$-th moment of $Y$ and the components of $X$ is necessary to ensure that the risk $R_p$ is finite for all $w \in \mathbb{R}^d$. The last assumption $\mathrm{E}[|Y - \langle w_p^*, X \rangle|^{2(p-2)}(X^j)^4] < \infty$ is a natural extension of the fourth moment assumption in Theorem 1. In fact, all three assumptions in Theorem 4 reduce to those of Theorem 1 as $p \to 2$. It is worth noting that the constant $c_p$ has the alternative expression $\sup_{w \in \mathbb{R}^d \setminus \{0\}}\{\|w\|_{L^p}/\|w\|_{H_p}\}$ by (4), i.e. it is the norm equivalence constant between the $L^p$ norm and the norm induced by $H_p$. Using again (4), we see that $c_p \to 1$ as $p \to 2$. As $p \to \infty$, $c_p$ grows, and we suspect in a dimension dependent way. However, this does not affect the asymptotic optimality of our rate as $c_p$ only enters an $o(1/n)$ term in our bound.

We now turn to the case of $p \in (1, 2)$ where we need a slightly stronger version of non-realizability.

**Theorem 5.** *Let $p \in (1, 2)$ and $\delta \in (0, 1]$. Assume that $\mathrm{P}(|\langle w_p^*, X \rangle - Y|^{2-p} > 0) = 1$ and $\mathrm{E}[|\langle w_p^*, X \rangle - Y|^{2(p-2)}] < \infty$. Further, assume that $\mathrm{E}[|Y|^p] < \infty$, $\mathrm{E}[(X^j)^2] < \infty$, $\mathrm{E}[|\langle w_p^*, X \rangle - Y|^{2(p-2)}(X^j)^4] < \infty$ for all $j \in [d]$. If*

$$n \geq 196\sigma_p^2(d + 2\log(4/\delta)),$$

*then, with probability at least $1 - \delta$*

$$R_p(\hat{w}_p) - R_p(w_p^*) \leq \frac{32768}{p-1} \frac{\mathrm{E}\left[\|\nabla \ell_p(\langle w_p^*, X \rangle - Y)\|_{H_p^{-1}}^2\right]}{n\delta}$$
$$+ \frac{1}{p-1}\left(\frac{2097152 \, \mathrm{E}\left[\|\nabla \ell_p(\langle w_p^*, X \rangle - Y)\|_{H_p^{-1}}^2\right]\sigma_p^{6-2p}c_p^{2-p}c_p^*}{n\delta}\right)^{1/(p-1)}$$

*where we used $c_p^*$ to denote $\mathrm{E}[|Y - \langle w_p^*, X \rangle|^{2(p-2)}]$ and $c_p := \mathrm{Tr}(H_p^{-1}\Sigma)$ where $\Sigma := \mathrm{E}[XX^T]$.*

Just like the bounds of Theorems 1 and 4, the bound of Theorem 5 is asymptotically optimal up to a constant factor that depends only on $p$. Indeed, since $1 < p < 2$, $1/(p-1) > 1$, and the second term is $o(1/n)$. At the level of assumptions, we have two additional conditions compared to Theorem 4. First, we require the existence of the second moment of the covariates instead of just the $p$-th moment. Second, we require a stronger version of non-realizability by assuming the existence of the $2(2-p)$ negative moment of $|\langle w_p^*, X \rangle - Y|$. In the majority of applications, an intercept variable is included as a covariate, i.e. $X^1 = 1$, so that this negative moment assumption is already implied by the standard assumption $\mathrm{E}[|Y - \langle w_p^*, X \rangle|^{2(p-2)}(X^j)^4] < \infty$. In the rare case where an intercept variable is not included, any negative moment assumption on $|\langle w_p^*, X \rangle - Y|$ can be used instead, at the cost of a larger factor in the $o(1/n)$ term.

Finally, it is worth noting that for the cases $p \in [1, 2)$, there are situations where the asymptotic bound (1) does not hold, as the limiting distribution of the coefficients $\hat{w}_p$ as $n \to \infty$ does not necessarily converge to a Gaussian, and depends heavily on the distribution of $\langle w_p^*, X \rangle - Y$, see e.g. Lai and Lee [LL05] and Knight [Kni98]. Overall, we suspect that perhaps a slightly weaker version of our assumptions is necessary for a fast rate like (1) to hold.

# 3 Proofs

## 3.1 Proof of Theorem 1

Here we give a detailed proof of Theorem 1. While the core technical result can be deduced by combining results from Oliveira [Oli16] and Lecué and Mendelson [LM16], here we frame the proof in a way that makes it easy to extend to the cases $p \in (1, \infty)$, and differently from either paper. We split the proof in three steps. First notice that since the loss is a quadratic function of $w$, we can express it exactly using a second order Taylor expansion around the minimizer $w_2^*$

$$\ell_2(\langle w, X \rangle - Y) - \ell_2(\langle w_2^*, X \rangle - Y) = \langle \nabla \ell_2(\langle w_2^*, X \rangle - Y), w - w_2^* \rangle + \frac{1}{2}\|w - w_2^*\|_{\nabla^2 \ell_2(\langle w_2^*, X \rangle - Y)}^2.$$

Taking empirical averages and expectations of both sides respectively shows that the excess empirical risk and excess risk also admit such an expansion

$$R_{2,n}(w) - R_{2,n}(w_2^*) = \langle \nabla R_{2,n}(w_2^*), w - w_2^* \rangle + \frac{1}{2}\|w - w_2^*\|_{H_{2,n}}^2,$$

$$R_2(w) - R_2(w_2^*) = \frac{1}{2}\|w - w_2^*\|_{H_2}^2, \tag{6}$$

where in the second equality we used that the gradient of the risk vanishes at the minimizer $w_2^*$. Therefore, to bound the excess risk, it is sufficient to bound the norm $\|w - w_2^*\|_{H_2}$. This is the goal of the second step, where we use two ideas. First, by definition, the excess empirical risk of the empirical risk minimizer satisfies the upper bound

$$R_{2,n}(\hat{w}_2) - R_{2,n}(w_2^*) \le 0. \tag{7}$$

Second, we use the Cauchy-Schwartz inequality to lower bound the excess empirical risk by

$$R_{2,n}(\hat{w}_2) - R_{2,n}(w_2^*) \ge -\|\nabla R_{2,n}(w_2^*)\|_{H_2^{-1}}\|\hat{w}_2 - w_2^*\|_{H_2} + \frac{1}{2}\|\hat{w}_2 - w_2^*\|_{H_{2,n}}^2, \tag{8}$$

and we further lower bound it by deriving high probability bounds on the two random terms $\|\nabla R_{2,n}(w_2^*)\|_{H_2^{-1}}$ and $\|\hat{w}_2 - w_2^*\|_{H_{2,n}}^2$. The first can easily be bounded using Chebyshev's inequality and the elementary fact that the variance of the average of $n$ i.i.d. random variables is the variance of their common distribution divided by $n$. Here we state the result for all $p \in (1, \infty)$; the straightforward proof can be found in the Appendix D.

**Lemma 2.** *Let* $p \in (1, \infty)$. *If* $p \in (1, 2)$, *let the assumptions of Theorem 5 hold. Then with probability at least* $1 - \delta/2$

$$\|\nabla R_{p,n}(w_p^*)\|_{H_p^{-1}} \le \sqrt{2 \operatorname{E}\left[\|\nabla \ell_p(\langle w_p^*, X \rangle - Y)\|_{H_p^{-1}}^2\right]/(n\delta)}.$$

For the second random term $\|\hat{w}_2 - w_2^*\|_{H_{2,n}}^2$, we use Theorem 3.1 of Oliveira [Oli16], which we restate here, emphasizing that the existence of fourth moments of the components of the random vector is enough to ensure the existence of the needed norm equivalence constant.

**Proposition 1** (Theorem 3.1, Oliveira [Oli16]). *Let* $Z \in \mathbb{R}^d$ *be a random vector satisfying* $\operatorname{E}[Z_j^4] < \infty$ *for all* $j \in [d]$ *and assume that* $\operatorname{P}(\langle v, Z \rangle = 0) = 1$ *only if* $v = 0$. *For* $p \in [1, \infty)$ *and* $v \in \mathbb{R}^d$, *define*

$$\|v\|_{L^p} := \operatorname{E}[(\langle v, Z \rangle)^p]^{1/p}, \qquad \sigma^2 := \left(\sup_{v \in \mathbb{R}^d \setminus \{0\}} \|v\|_{L^4}/\|v\|_{L^2}\right)^4.$$

*Let* $(Z_i)_{i=1}^n$ *be i.i.d. samples of* $Z$. *Then, with probability at least* $1 - \delta$, *for all* $v \in \mathbb{R}^d$,

$$\frac{1}{n}\sum_{i=1}^n \langle v, Z_i \rangle^2 \ge \left(1 - 7\sigma\sqrt{\frac{d + 2\log(2/\delta)}{n}}\right)\|v\|_{L^2}^2.$$

Using this result we can immediately deduce the required high probability lower bound on the second random term $\|\hat{w}_2 - w_2^*\|_{H_{2,n}}^2$; we leave the obvious proof to Appendix D.

**Corollary 1.** *Under the assumptions of Theorem 1, if $n \geq 196\sigma_2^2(d + 2\log(4/\delta))$, then with probability at least $1 - \delta/2$, for all $w \in \mathbb{R}^d$,*

$$\|w - w_2^*\|_{H_{2,n}}^2 \geq \frac{1}{2}\|w - w_2^*\|_{H_2}^2.$$

Combining Lemma 2, Corollary 1, and (8) yields that with probability at least $1 - \delta$

$$R_{2,n}(\hat{w}_2) - R_{2,n}(w_2^*) \geq -\sqrt{2\,\mathrm{E}\Big[\|\nabla\ell_p(\langle w_p^*, X\rangle - Y)\|_{H_p^{-1}}^2\Big]/(n\delta)}\ \|\hat{w}_2 - w_2^*\|_{H_2} + \frac{1}{4}\|\hat{w}_2 - w_2^*\|_{H_2}^2.$$

$$(9)$$

Finally, combining (7) and (9) gives that with probability at least $1 - \delta$

$$\|\hat{w}_2 - w_2^*\|_{H_2} \leq 4\sqrt{2\,\mathrm{E}\Big[\|\nabla\ell_p(\langle w_p^*, X\rangle - Y)\|_{H_p^{-1}}^2\Big]/(n\delta)}.$$

Replacing in (6) finishes the proof. □

## 3.2 Proof Sketch of Theorem 4

The main challenge in moving from the case $p = 2$ to the case $p \in (2, \infty)$ is that the second order Taylor expansion of the loss is no longer exact. The standard way to deal with this problem is to assume that the loss is upper and lower bounded by quadratic functions, i.e. that it is smooth and strongly convex. Unfortunately, as discussed in Section 1, the $\ell_p$ loss is not strongly convex for any $p > 2$, so we need to find another way to deal with this issue. Once this has been resolved however, the strategy we used in the proof of Theorem 1 can be applied almost verbatim to yield the result. Remarkably, a result of [AKPS22] allows us to upper and lower bound the $p$-th power loss for $p \in (2, \infty)$ by its second order Taylor expansion around a point, up to some residual terms. An application of this result yields the following Lemma.

**Lemma 3.** *Let $p \in (2, \infty)$. Then:*

$$R_{p,n}(w) - R_{p,n}(w_p^*) \geq \frac{1}{8(p-1)}\|w - w_p^*\|_{H_{p,n}}^2 + \langle \nabla R_{p,n}(w_p^*), w - w_p^*\rangle, \qquad (10)$$

$$R_p(w) - R_p(w_p^*) \leq \frac{2p}{(p-1)}\|w - w_p^*\|_{H_p}^2 + p^p\|w - w_p^*\|_{L^p}^p. \qquad (11)$$

Up to constant factors that depend only on $p$ and an $L^p$ norm residual term, Lemma 3 gives matching upper and lower bounds on the excess risk and excess empirical risk in terms of their second order Taylor expansions around the minimizer. We can thus use the approach taken in the proof of Theorem 1 to obtain the result. The only additional challenge is the control of the term $\|\hat{w}_p - w_p^*\|_{L^p}$, which we achieve by reducing it to an $\|\hat{w}_p - w_p^*\|_{H_p}$ term using norm equivalence. A detailed proof of Theorem 4, including the proof of Lemma 3, can be found in Appendix E.

## 3.3 Proof Sketch of Theorem 5

The most technically challenging case is when $p \in (1, 2)$. Indeed as seen in the proof of Theorem 1, the most involved step is lower bounding the excess empirical risk with high probability. For the case $p \in [2, \infty)$, we achieved this by having access to a pointwise quadratic lower bound, which is not too surprising. Indeed, at small scales, we expect the second order Taylor expansion to be accurate, while at large scales, we expect the $p$-th power loss to grow at least quadratically for $p \in [2, \infty)$.

In the case of $p \in (1, 2)$, we are faced with a harder problem. Indeed, as $p \to 1$, the $\ell_p$ losses behave almost linearly at large scales. This means that we cannot expect to obtain a global quadratic lower bound as for the case $p \in [2, \infty)$, so we will need a different proof technique. Motivated by related concerns, Bubeck, Cohen, Lee, and Li [BCLL18] introduced the following approximation to the $p$-th power function

$$\gamma_p(t, x) := \begin{cases} \dfrac{p}{2}t^{p-2}x^2 & \text{if} \quad x \leq t \\ x^p - \left(1 - \dfrac{p}{2}\right)t^p & \text{if} \quad x > t, \end{cases}$$

for $t, x \in [0, \infty)$ and with $\gamma_p(0,0) = 0$. This function was further studied by Adil, Kyng, Peng, and Sachdeva [AKPS19], who showed in particular that for any $t \in \mathbb{R}$, the function $x \mapsto \gamma_p(|t|, |x|)$ is, up to constants that depend only on $p$, equal to the gap between the function $x \mapsto \ell_p(t + x)$ and its linearization around 0; see Lemma 4.5 in [AKPS19] for the precise statement. We use this result to derive the following Lemma.

**Lemma 4.** *Let $p \in (1, 2)$. Under the assumptions of Theorem 5, we have*

$$R_{p,n}(w) - R_{p,n}(w_p^*) \geq \frac{1}{4p^2} \frac{1}{n} \sum_{i=1}^{n} \gamma_p\big(|\langle w_p^*, X_i \rangle - Y_i|, |\langle w - w_p^*, X_i \rangle|\big) + \langle \nabla R_{p,n}(w_p^*), w - w_p^* \rangle,$$
$$(12)$$

$$R_p(w) - R_p(w_p^*) \leq \frac{4}{(p-1)} \|w - w_p^*\|_{H_p}^2.$$
$$(13)$$

As expected, while we do have the desired quadratic upper bound, the lower bound is much more cumbersome, and is only comparable to the second order Taylor expansion when $|\langle w - w_p^*, X_i \rangle| \leq |\langle w_p^*, X_i \rangle - Y_i|$. What we need for the proof to go through is a high probability lower bound of order $\Omega(\|w - w^*\|_{H_p}^2)$ on the first term in the lower bound (12). We obtain this in the following Proposition.

**Proposition 2.** *Let $\delta \in (0, 1]$. Under the assumptions of Theorem 5, if $n \geq 196\sigma_p^2(d + 2\log(4/\delta))$, then with probability at least $1 - \delta/2$, for all $w \in \mathbb{R}^d$,*

$$\frac{1}{n} \sum_{i=1}^{n} \gamma_p\big(|\langle w_p^*, X_i \rangle - Y_i|, |\langle w - w_p^*, X_i \rangle|\big) \geq \frac{p}{8} \min\Big\{ \|w - w_p^*\|_{H_p}^2, \varepsilon^{2-p}\|w - w_p^*\|_{H_p}^p \Big\},$$

*where $\varepsilon^{p-2} := 8\sigma_p^{3-p} c_p^{(2-p)/2} \sqrt{c_p^*}$, and $c_p$ and $c_p^*$ are as defined in Theorem 5.*

*Proof.* Let $\varepsilon > 0$ and let $T \in (0, \infty)$ be a truncation parameter we will set later. Define

$$\tilde{X} := X \cdot \mathbb{1}_{[0,T]}(\|X\|_{H_p^{-1}}),$$

and the constant $\beta := T\varepsilon$. By Lemma 3.3 in [AKPS19], we have that $\gamma_p(t, \lambda x) \geq \min\{\lambda^2, \lambda^p\}\gamma_p(t, x)$ for all $\lambda \geq 0$. Furthermore, it is straightforward to verify that $\gamma_p(t, x)$ is decreasing in $t$ and increasing in $x$. Therefore, we have, for all $w \in \mathbb{R}^d$,

$$\frac{1}{n} \sum_{i=1}^{n} \gamma_p\big(|\langle w_p^*, X_i \rangle - Y_i|, |\langle w - w_p^*, X_i \rangle|\big)$$

$$\geq \min\Big\{ \varepsilon^{-2}\|w - w_p\|_{H_p}^2, \varepsilon^{-p}\|w - w_p\|_{H_p}^p \Big\} \frac{1}{n} \sum_{i=1}^{n} \gamma_p\left( |\langle w_p^*, X_i \rangle - Y_i|, \left| \left\langle \frac{\varepsilon(w - w_p^*)}{\|w - w_p^*\|_{H_p}}, X_i \right\rangle \right| \right)$$

$$\geq \min\Big\{ \varepsilon^{-2}\|w - w_p\|_{H_p}^2, \varepsilon^{-p}\|w - w_p\|_{H_p}^p \Big\} \cdot \inf_{\|w\|_{H_p} = \varepsilon} \frac{1}{n} \sum_{i=1}^{n} \gamma_p\big(|\langle w_p^*, X_i \rangle - Y_i|, |\langle w, X_i \rangle|\big).$$
$$(14)$$

The key idea to control the infimum in (14) is to truncate $\langle w, X_i \rangle$ from above by using the truncated vector $\tilde{X}$, and $|\langle w_p^*, X_i \rangle - Y_i|$ from below by forcing it to be greater than $\beta$. By the monotonicity properties of $\gamma_p$ discussed above, we get that

$$\inf_{\|w\|_{H_p} = \varepsilon} \frac{1}{n} \sum_{i=1}^{n} \gamma_p\big(|\langle w_p^*, X_i \rangle - Y_i|, |\langle w, X_i \rangle|\big)$$

$$\geq \inf_{\|w\|_{H_p} = \varepsilon} \frac{1}{n} \sum_{i=1}^{n} \gamma_p(\max\{|\langle w_p^*, X_i \rangle - Y_i|, \beta\}, |\langle w, \tilde{X}_i \rangle|)$$

$$= \frac{\varepsilon^2 p}{2} \inf_{\|w\|_{H_p} = 1} \frac{1}{n} \sum_{i=1}^{n} \max\{|\langle w_p^*, X_i \rangle - Y_i|, \beta\}^{p-2} |\langle w, \tilde{X}_i \rangle|^2,$$
$$(15)$$

where the equality follows by the fact that with the chosen truncations, the second argument of $\gamma_p$ is less than or equal to the first. It remains to lower bound the infimum in (15). Define

$$Z = \max\{|\langle w_p^*, X \rangle - Y|, \beta\}^{(p-2)/2} \tilde{X}.$$

Removing the truncations and using our assumptions, we see that the components of $Z$ have finite fourth moment. By Proposition 1 and the condition on $n$, we get that with probability at least $1 - \delta/2$,

$$\inf_{\|w\|_{H_p}=1} \frac{1}{n} \sum_{i=1}^n \max\{|\langle w_p^*, X_i \rangle - Y_i|, \beta\}^{p-2} |\langle w, \tilde{X}_i \rangle|^2$$

$$= \inf_{\|w\|_{H_p}=1} \frac{1}{n} \sum_{i=1}^n \langle w, Z_i \rangle^2 \geq \frac{1}{2} \inf_{\|w\|_{H_p}=1} \mathrm{E}\big[\langle w, Z \rangle^2\big]$$

$$= \frac{1}{2} \inf_{\|w\|_{H_p}=1} \mathrm{E}\Big[\max\{|\langle w_p^*, X \rangle - Y|, \beta\}^{(p-2)} \langle w, \tilde{X} \rangle^2\Big]$$

$$\geq \frac{1}{2}\left(1 - \sup_{\|w\|_{H_p}=1} \mathrm{E}\Big[|\langle w_p^*, X \rangle - Y|^{p-2} \langle w, X \rangle^2 \Big(\mathbb{1}_{[0,\beta)}(|\langle w_p^*, X \rangle - Y|) + \mathbb{1}_{(T,\infty)}(\|X\|_{H_p^{-1}})\Big)\Big]\right)$$

$$\tag{16}$$

We now bound the supremum in (16). We have

$$\sup_{\|w\|_{H_p}=1} \mathrm{E}\Big[|\langle w_p^*, X \rangle - Y|^{p-2} \langle w, X \rangle^2 \Big(\mathbb{1}_{[0,\beta)}(|\langle w_p^*, X \rangle - Y|) + \mathbb{1}_{(T,\infty)}(\|X\|_{H_p^{-1}})\Big)\Big]$$

$$\leq \sup_{\|w\|_{H_p}=1} \left\{\mathrm{E}\Big[|\langle w_p^*, X \rangle - Y|^{2(p-2)} \langle w, X \rangle^4\Big]^{1/2}\right\} \Big(\mathrm{P}\big(|\langle w_p^*, X \rangle - Y| < \beta\big) + \mathrm{P}\Big(\|X\|_{H_p^{-1}} > T\Big)\Big)^{1/2}$$

$$= \left(\sup_{\|w\|_{H_p}=1} \|w\|_{L^4,p}^2\right) \Big(\mathrm{P}\big(|\langle w_p^*, X \rangle - Y| < \beta\big) + \mathrm{P}\Big(\|X\|_{H_p^{-1}} > T\Big)\Big)^{1/2}$$

$$= \sigma_p \Big(\mathrm{P}\big(|\langle w_p^*, X \rangle - Y| < \beta\big) + \mathrm{P}\Big(\|X\|_{H_p^{-1}} > T\Big)\Big)^{1/2}, \tag{17}$$

where the first inequality follows from Cauchy-Schwartz inequality, and the subsequent equalities by definitions of $\|\cdot\|_{L^4,p}$ and $\sigma_p^2$. It remains to bound the tail probabilities. Recall that $\beta = T\varepsilon$, so

$$\mathrm{P}\big(|\langle w_p^*, X \rangle - Y| < \beta\big) = \mathrm{P}\big(|\langle w_p^*, X \rangle - Y| < T\varepsilon\big)$$

$$= \mathrm{P}\big(|\langle w_p^*, X \rangle - Y|^{-1} > (T\varepsilon)^{-1}\big)$$

$$= \mathrm{P}\Big(|\langle w_p^*, X \rangle - Y|^{2(p-2)} > (T\varepsilon)^{2(p-2)}\Big)$$

$$\leq \mathrm{E}[|\langle w_p^*, X \rangle - Y|^{2(p-2)}](T\varepsilon)^{2(2-p)}$$

$$= c_p^*(T\varepsilon)^{2(2-p)},$$

where we applied Markov's inequality in the fourth line, and the last follows by definition of $c_p^*$ in Theorem 5. Moreover by the finiteness of the second moment of the coordinates $X^j$ of $X$, we have

$$\mathrm{E}\Big[\|X\|_{H_p^{-1}}^2\Big] = \mathrm{E}\big[X^T H_p^{-1} X\big] = \mathrm{E}\big[\mathrm{Tr}\big(H_p^{-1} X X^T\big)\big] = \mathrm{Tr}\big(H_p^{-1} \Sigma\big) = c_p$$

where $\Sigma = \mathrm{E}[X X^T]$, and the last equality by definition of $c_p$ in Theorem 5. By Markov's inequality

$$\mathrm{P}\big(|\langle w_p^*, X \rangle - Y| < T\varepsilon\big) + \mathrm{P}\Big(\|X\|_{H_p^{-1}} > T\Big) \leq c_p^* T^{2(2-p)} \varepsilon^{2(2-p)} + \frac{c_p}{T^2}.$$

Choosing

$$T := \left(\frac{c_p}{c_p^*(2-p)}\right)^{1/(6-2p)}, \qquad \varepsilon^{2-p} := \frac{1}{8\sigma_p^{3-p} \sqrt{c_p^*} \cdot c_p^{(2-p)/2}},$$

ensures that

$$\sigma_p\Big(\mathrm{P}\big(|\langle w_p^*, X \rangle - Y| < T^*\varepsilon\big) + \mathrm{P}\Big(\|X\|_{H_p^{-1}} > T^*\Big)\Big)^{1/2} \leq 1/2. \tag{18}$$

Combining the inequalities (18), (17), (16), (15), and (14) yields the result. $\qquad\square$

A detailed proof of Theorem 5, including the proof of Lemma 4, can be found in Appendix F.

## Acknowledgments and Disclosure of Funding

We thank Nikita Zhivotovskiy, Sushant Sachdeva, and Deeksha Adil for feedback on the manuscript. MAE was partially supported by NSERC Grant [2019-06167], CIFAR AI Chairs program, and CIFAR AI Catalyst grant.

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

# A Differentiability of the risk

In this section, we rigorously establish the twice differentiability the risk under our assumptions. We start by showing that under a subset of our assumptions, the risk is differentiable everywhere on $\mathbb{R}^d$.

**Lemma 5.** *Let $p \in (1, \infty)$ and assume that $\mathrm{E}[|Y|^p] < \infty$ and $\mathrm{E}[|X_j|^p] < \infty$ for all $j \in [d]$. Then $R_p$ is differentiable on $\mathbb{R}^d$, and*

$$\nabla R_p(w) = \mathrm{E}[\nabla \ell_p(\langle w, X \rangle - Y)].$$

*Proof.* Let $w \in \mathbb{R}^d$. We want to show that

$$\lim_{\Delta \to 0} \frac{|R_p(w + \Delta) - R_p(w) - \mathrm{E}[\langle \nabla \ell_p(\langle w, X \rangle - Y), \Delta \rangle]|}{\|\Delta\|} = 0,$$

where, for convenience, we take the norm $\|\cdot\|$ to be the Euclidean norm. Define the function $\phi(w, X, Y) := \ell_p(\langle w, X \rangle - Y)$ and note that by the chain rule $\phi$ is differentiable as a function of $w$ on all of $\mathbb{R}^d$. Now let $(\Delta_k)_{k=1}^n$ be a sequence in $\mathbb{R}^d$ such that $\lim_{k \to \infty} \|\Delta_k\| = 0$. Then

$$
\begin{aligned}
&\lim_{k \to \infty} \frac{|R_p(w + \Delta_k) - R_p(w) - \mathrm{E}[\langle \nabla \phi(w, X, Y), \Delta_k \rangle]|}{\|\Delta_k\|} \\
&= \lim_{k \to \infty} \frac{|\mathrm{E}[\phi(w + \Delta_k, X, Y) - \phi(w, X, Y) - \langle \nabla \phi(w, X, Y), \Delta_k \rangle]|}{\|\Delta_k\|} \\
&\le \lim_{k \to \infty} \mathrm{E}\left[ \frac{|\phi(w + \Delta_k, X, Y) - \phi(w, X, Y) - \langle \nabla \phi(w, X, Y), \Delta_k \rangle|}{\|\Delta_k\|} \right].
\end{aligned}
\tag{19}
$$

Our goal is to interchange the limit and expectation. For that, we will use the dominated convergence theorem. We construct our dominating function as follows. Let $R := \sup_{k \in \mathbb{N}} \|\Delta_k\|$, and note that $R < \infty$ since $\|\Delta_k\| \to 0$ as $k \to \infty$. Then we have

$$
\begin{aligned}
&\frac{|\phi(w + \Delta_k, X, Y) - \phi(w, X, Y) - \langle \nabla \phi(w, X, Y), \Delta_k \rangle|}{\|\Delta_k\|} \\
&\le \frac{|\phi(w + \Delta_k, X, Y) - \phi(w, X, Y)|}{\|\Delta_k\|} + \frac{|\langle \nabla \phi(w, X, Y), \Delta_k \rangle|}{\|\Delta_k\|} \\
&\le \frac{\left\langle \int_0^1 \nabla \phi(w + t\Delta_k, X, Y) dt, \Delta_k \right\rangle}{\|\Delta_k\|} + \|\nabla \phi(w, X, Y)\| \\
&\le \left\| \int_0^1 \nabla \phi(w + t\Delta_k, X, Y) dt \right\| + \|\nabla \phi(w, X, Y)\| \\
&\le \int_0^1 \|\nabla \phi(w + t\Delta_k, X, Y)\| dt + \|\nabla \phi(w, X, Y)\| \\
&\le 2 \sup_{\Delta \in B(0, R)} \|\nabla \phi(w + \Delta, X, Y)\| \\
&\le \frac{2}{p-1} \|X\| \sup_{\Delta \in B(0, R)} |\langle w + \Delta, X \rangle - Y|^{p-1} \\
&\le \frac{2}{p-1} \|X\| \sup_{\Delta \in B(0, R)} \max\{2^{p-1}, 1\} \left( |\langle w, X \rangle - Y|^{p-1} + |\langle \Delta, X \rangle|^{p-1} \right) \\
&= \frac{2^p}{p-1} \left\{ |\langle w, X \rangle - Y|^{p-1} \|X\| + R^{p-1} \|X\|^p \right\} =: g(X, Y),
\end{aligned}
$$

where the second line follows by triangle inequality, the third from the fundamental theorem of calculus applied component-wise, the fourth by Cauchy-Schwartz inequality, the fifth by Jensen's inequality and the convexity of the norm, and the eighth by the inequality $|a + b|^q \le \max\{2^{q-1}, 1\}(|a|^q + |b|^q)$

valid for $q > 0$. It remains to show that $g(X, Y)$ is integrable. We have

$$
\begin{aligned}
\mathrm{E}[g(X, Y)] &= \frac{2^p}{p-1} \mathrm{E}\Big[|\langle w, X\rangle - Y|^{p-1}\|X\| + R^{p-1}\|X\|^p\Big] \\
&= \frac{2^p}{p-1}\left\{\sum_{j=1}^d \mathrm{E}\Big[|\langle w, X\rangle - Y|^{p-1}|X^j|\Big] + R^{p-1}\mathrm{E}\left[\left(\sum_{j=1}^d|X^j|\right)^p\right]\right\} \\
&\leq \frac{2^p}{p-1}\left\{\sum_{j=1}^d \mathrm{E}[|\langle w, X\rangle - Y|^p]^{\frac{p-1}{p}}\mathrm{E}[|X_j|^p]^{1/p} + R^{p-1}d^p\sum_{j=1}^d \mathrm{E}\Big[|X^j|^p\Big]\right\} \\
&< \infty,
\end{aligned}
$$

where in the second line we used that the Euclidean norm is bounded by the 1-norm, in the third we used Holder's inequality, and the last line follows from our assumptions. Applying the dominated convergence theorem, we interchange the limit and the expectation in (19). Recalling that $\phi$ is differentiable finishes the proof. □

We now turn to the twice differentiability of the risk. We start with the easy case $p \in [2, \infty)$. The proof is very similar to that of Lemma 5 and we omit it here.

**Lemma 6.** *Let $p \in [2, \infty)$ and assume that $\mathrm{E}[|Y|^p] < \infty$ and $\mathrm{E}[|X_j|^p] < \infty$ for all $j \in [d]$. Then $R_p$ is twice differentiable on $\mathbb{R}^d$, and*

$$
\nabla^2 R_p(w) = \mathrm{E}[\nabla^2 \ell_p(\langle w, X\rangle - Y)].
$$

The case $p \in (1, 2)$ is more complicated. The following lemma establishes the twice differentiability of the risk at its minimizer under a subset of the assumptions of Theorem 5.

**Lemma 7.** *Let $p \in (1, 2)$. Assume that $\mathrm{P}\big(|\langle w_p^*, X\rangle - Y| = 0\big) = 0$ and $\mathrm{E}[|\langle w_p^*, X\rangle - Y|^{p-2}(X^j)^2] < \infty$ for all $j \in [d]$. Then $R_p$ is twice differentiable at $w_p^*$ and*

$$
\nabla^2 R_p(w_p^*) = \mathrm{E}[\nabla^2 \ell_p(\langle w_p^*, X\rangle - Y)]
$$

*Proof.* The difficulty in the proof compared to Lemma 5 and Lemma 6 stems from the fact that the loss is not twice differentiable at zero. We still rely on the dominated convergence theorem, but the construction of the dominating function is slightly more intricate. Using the setup of the proof of Lemma 5, and following the same line of arguments, we arrive at

$$
\begin{aligned}
&\lim_{k\to\infty} \frac{\|\nabla R_p(w_p^* + \Delta_k) - \nabla R_p(w_p^*) - \mathrm{E}\big[\nabla^2\phi(w_p^*, X, Y)\Delta_k\big]\|}{\|\Delta_k\|} \\
&\leq \lim_{k\to\infty} \mathrm{E}\left[\frac{\|\nabla\phi(w_p^* + \Delta_k, X, Y) - \nabla\phi(w_p^*, X, Y) - \nabla^2\phi(w_p^*, X, Y)\Delta_k\|}{\|\Delta_k\|}\right],
\end{aligned}
\tag{20}
$$

where we have used the fact that since $\mathrm{P}\big(|\langle w_p^*, X\rangle - Y| = 0\big) = 0$, $\phi(w, X, Y)$ is almost surely twice differentiable at $w_p^*$. To finish the proof, it remains to construct a dominating function for the above sequence to justify the interchange of the limit and expectation. We consider two cases.

**Case 1:** $\|\Delta_k\| \geq \big|\langle w_p^*, X\rangle - Y\big|/(2\|X\|) =: R(X,Y)$. Then we have

$$\frac{\|\nabla\phi(w_p^* + \Delta_k, X, Y) - \nabla\phi(w_p^*, X, Y) - \nabla^2\phi(w_p^*, X, Y)\Delta_k\|}{\|\Delta_k\|}$$

$$\leq \frac{\|\nabla\phi(w_p^* + \Delta_k, X, Y)\| + \|\nabla\phi(w_p^*, X, Y)\| + \|\nabla^2\phi(w_p^*, X, Y)\Delta_k\|}{\|\Delta_k\|}$$

$$\leq \frac{\left(\big|\langle w_p^* + \Delta, X\rangle - Y\big|^{p-1} + \big|\langle w_p^*, X\rangle - Y\big|^{p-1}\right)\|X\|}{(p-1)\|\Delta_k\|} + \|\nabla^2\phi(w_p^*, X, Y)\|_{op}$$

$$\leq \frac{2\big|\langle w_p^*, X\rangle - Y\big|^{p-1}\|X\|}{(p-1)\|\Delta_k\|} + \big|\langle w_p^*, X\rangle - Y\big|^{p-2}\|X\|^2 + \frac{\big|\langle \Delta_k/\|\Delta_k\|, X\rangle\big|^{p-1}\|X\|}{(p-1)\|\Delta_k\|^{2-p}}$$

$$\leq \frac{4\big|\langle w_p^*, X\rangle - Y\big|^{p-2}\|X\|^2}{(p-1)} + \big|\langle w_p^*, X\rangle - Y\big|^{p-2}\|X\|^2 + \frac{\|X\|^p}{(p-1)\|\Delta_k\|^{2-p}}$$

$$\leq \frac{7\big|\langle w_p^*, X\rangle - Y\big|^{p-2}\|X\|^2}{(p-1)}$$

where the second line follows by triangle inequality, the third by definition of the operator norm, the fourth by $|a + b|^q \leq |a|^q + |b|^q$ valid for $q \in (0,1)$, and the fifth and sixth by Cauchy-Schwartz inequality and the assumed lower bound on $\|\Delta_k\|$.

**Case 2:** $\|\Delta_k\| < R(X,Y)$. We start by noting that, for all $\Delta \in B(0, R(X,Y)) := \{x \in \mathbb{R}^d \mid \|x\| < R(X,Y)\}$, we have

$$\big|\langle w_p^* + \Delta, X\rangle - Y\big| \geq \big|\langle w_p^*, X\rangle - Y\big| - |\langle \Delta, X\rangle| \geq \big|\langle w_p^*, X\rangle - Y\big| - \|\Delta\|\|X\| > \big|\langle w_p^*, X\rangle - Y\big|/2 > 0.$$

Therefore $\phi(w, X, Y)$ is twice differentiable on $B(0, R(X,Y))$. Now

$$\frac{\|\nabla\phi(w_p^* + \Delta_k, X, Y) - \nabla\phi(w_p^*, X, Y) - \nabla^2\phi(w_p^*, X, Y)\Delta_k\|}{\|\Delta_k\|}$$

$$\leq \frac{\|\nabla\phi(w_p^* + \Delta_k, X, Y) - \nabla\phi(w_p^*, X, Y)\| + \|\nabla^2\phi(w_p^*, X, Y)\Delta_k\|}{\|\Delta_k\|}$$

$$\leq \frac{\left\|\left(\int_0^1 \nabla^2\phi(w_p^* + t\Delta_k, X, Y)dt\right)\Delta_k\right\|}{\|\Delta_k\|} + \|\nabla^2\phi(w_p^*, X, Y)\|_{op}$$

$$\leq \left\|\int_0^1 \nabla^2\phi(w + t\Delta_k, X, Y)dt\right\|_{op} + \|\nabla^2\phi(w_p^*, X, Y)\|_{op}$$

$$\leq \int_0^1 \|\nabla^2\phi(w + t\Delta_k, X, Y)\|_{op}dt + \|\nabla^2\phi(w_p^*, X, Y)\|_{op}$$

$$\leq 2 \sup_{\Delta \in B(0, R(X,Y))} \|\nabla^2\phi(w_p^* + \Delta, X, Y)\|_{op}$$

$$\leq 2\|X\|_2^2 \sup_{\Delta \in B(0, R(X,Y))} \big|\langle w_p^* + \Delta, X\rangle - Y\big|^{p-2}$$

$$\leq 4\big|\langle w_p^*, X\rangle - Y\big|^{p-2}\|X\|_2^2$$

where the second line follows from the triangle inequality, the third follows from the twice differentiability of $\phi$ on $B(0, R(X,Y))$ and the fundamental theorem of calculus applied component-wise, the fifth by Jensen's inequality, and the last by definition of $R(X,Y)$ and the above lower bound. We therefore define our dominating function by

$$g(X, Y) := 8\big|\langle w_p^*, X\rangle - Y\big|^{p-2}\|X\|_2^2.$$

It is then immediate from our assumptions that $g(X, Y)$ is integrable. Interchanging the limit and the expectation in (20) and recalling that $\phi$ is almost surely twice differentiable at $w_p^*$ finishes the proof.

$\square$

# B Proof of Lemma 1

We start with the claim that $\rho_0$ is upper-semicontinuous. We want to show that for any $w \in \mathbb{R}^d$ and any sequence $(w_k)_{k=1}^\infty$ converging to $w$ (in the norm topology)

$$\limsup_{k \to \infty} \rho_0(w_k) \leq \rho_0(w).$$

Fix a $w \in \mathbb{R}^d$ and let $(w_k)_{k=1}^\infty$ be a sequence in $\mathbb{R}^d$ satisfying $\lim_{k \to \infty} \|w - w_k\| = 0$, where for convenience we take $\|\cdot\|$ to be the Euclidean norm on $\mathbb{R}^d$. Then we have by (reverse) Fatou's Lemma

$$\limsup_{k \to \infty} \rho_0(w_k) = \limsup_{k \to \infty} \mathrm{E}\big[\mathbb{1}_{\{0\}}(\langle w_k, X \rangle)\big] \leq \mathrm{E}\Big[\limsup_{k \to \infty} \mathbb{1}_{\{0\}}(\langle w_k, X \rangle)\Big]. \tag{21}$$

Now we bound the inner limsup pointwise. We split this task in two cases. If $\langle w, X \rangle = 0$, then

$$\limsup_{k \to \infty} \mathbb{1}_{\{0\}}(\langle w_k, X \rangle) \leq 1 = \mathbb{1}_{\{0\}}(\langle w, X \rangle). \tag{22}$$

Otherwise we have $\delta := |\langle w, X \rangle| > 0$. But then, by the convergence of $(w_k)_{k=1}^\infty$ to $w$, there exists a $K \in \mathbb{N}$ such that for all $k \geq K$ we have $\|w_k - w\| < \delta/(2\|X\|)$. This implies that for all $k \geq K$

$$|\langle w_k, X \rangle| = |\langle w, X \rangle - \langle w - w_k, X \rangle| \geq |\langle w, X \rangle| - |\langle w - w_k, X \rangle| \geq \delta - \|w_k - w\|_2 \|X\| \geq \delta/2 > 0.$$

We conclude that

$$\limsup_{k \to \infty} \mathbb{1}_{\{0\}}(\langle w_k, X \rangle) = \lim_{k \to \infty} \mathbb{1}_{\{0\}}(\langle w_k, X \rangle) = 0 = \mathbb{1}_{\{0\}}(\langle w, X \rangle). \tag{23}$$

Combining (21), (22), and (23) proves the upper-semicontinuity of $\rho_0$. Essentially the same proof shows the lower-semicontinuity of $\rho_q(\cdot, \kappa)$ for any $\kappa \geq 0$; we omit it here.

For the remaining claims, first notice that the function $\rho_0$ is scale invariant, i.e. for all $w \in \mathbb{R}^d$ and all $c \in \mathbb{R}$, we have $\rho_0(cw) = \rho_0(w)$. Therefore

$$\sup_{w \in \mathbb{R}^d \setminus \{0\}} \rho_0(w) = \sup_{w \in \mathbb{S}^{d-1}} \rho_0(w),$$

where $\mathbb{S}^{d-1}$ is the Euclidean unit sphere. By assumption on the random vector $X$, we know that $\rho_0(w) < 1$ for all $w \in \mathbb{S}^{d-1}$. Furthermore since $\rho_0$ is upper semicontinuous, and $\mathbb{S}^{d-1}$ is compact, $\rho_0$ attains its supremum on $\mathbb{S}^{d-1}$ at some point $w_0 \in \mathbb{S}^{d-1}$. From this we conclude that

$$\rho = \sup_{w \in \mathbb{R}^d \setminus \{0\}} \rho_0(w) = \rho_0(w_0) < 1.$$

Finally, we turn to the claim about $\rho_q$. Since $\mathrm{E}[|X^j|^q] < \infty$, the function $\|\cdot\|_{L^q}$ is a norm on $\mathbb{R}^d$, from which it follows that $\rho_q(w, \kappa)$ is also scale invariant for any $\kappa$. Therefore

$$\inf_{w \in \mathbb{R}^d \setminus \{0\}} \rho_q(w, \kappa) = \inf_{w \in S_q} \rho_q(w, \kappa),$$

where $S_q$ is the unit sphere of the norm $\|\cdot\|_{L^q}$. Now fix $\kappa \in [0, 1)$. We claim that $\rho_q(w, \kappa) > 0$ for all $w \in S_q$. Suppose not. Then there exists a $w \in S_q$ such that $|\langle w, X \rangle| \leq \kappa$ with probability 1, but then we get the contradiction

$$1 = \|w\|_{L^q} = \mathrm{E}[|\langle w, X \rangle|^q]^{1/q} \leq \kappa < 1.$$

therefore $\rho_q(w, \kappa) > 0$ for all $w \in S_q$. Now since $\rho_q(\cdot, \kappa)$ is lower-semicontinuous, and $S_q$ is compact, $\rho_q(\cdot, \kappa)$ attains its infimum on $S_q$ at some point $w_q \in S_q$. From this we conclude

$$\inf_{w \in \mathbb{R}^d \setminus \{0\}} \rho_q(w, \kappa) = \rho_q(w_q, \kappa) > 0.$$

$\square$

# C Proof of Theorem 3

Fix $p \in (1, \infty)$, and let $\hat{w} := \hat{w}_p$. Our goal will be to upper bound the probability $P(\hat{w} \neq w^*)$. By assumption, we have that $Y = \langle w^*, X \rangle$, so that $Y_i = \langle w^*, X_i \rangle$ for all $i \in [n]$. Since $\hat{w}$ minimizes the empirical risk, we must also have $\langle \hat{w}, X_i \rangle = Y_i = \langle w^*, X_i \rangle$ for all $i \in [n]$. Let $A \in \mathbb{R}^{n \times d}$ denote the matrix whose $i$-th row is $X_i$. Then we have the following implications.

$$\hat{w} \neq w^* \Rightarrow \langle \hat{w} - w^*, X_i \rangle = 0 \ \forall i \in [n] \Rightarrow \exists w \in \mathbb{R}^d \setminus \{0\} \mid Aw = 0 \Leftrightarrow \text{rowrank}(A) < d. \quad (24)$$

Let $r := \text{rowrank}(A)$. We claim the following equivalence

$$\text{rowrank}(A) < d \Leftrightarrow \exists S \subset [n] \mid |S| = d - 1 \wedge \forall i \in [n] \setminus S \ X_i \in \text{span}(\{X_k \mid k \in S\}). \quad (25)$$

Indeed the implication ($\Leftarrow$) follows by definition of the rowrank of $A$. For the implication ($\Rightarrow$), by definition, $\{X_i \mid i \in [n]\}$ is a spanning set for the row space of $A$, therefore it can be reduced to a basis of it $\{X_k \mid k \in S_1\}$ for some indices $S_1 \subset [n]$ with $|S_1| = r$. If $r = d - 1$, then the choice $S = S_1$ satisfies the right side of (25). Otherwise, let $S_2 \subset [n] \setminus S_1$ with $|S_2| = d - 1 - r$. Such a subset exists since by assumption $n \geq d > d - 1$. Then the set $S := S_1 \cup S_2$ satisfies the right side of (25). Combining (24) and (25) we arrive at:

$$P(\hat{w} \neq w^*) \leq P\left( \bigcup_{\substack{S \subset [n] \\ |S| = d-1}} \{\forall i \in [n] \setminus S \ X_i \in \text{span}(\{X_k \mid k \in S\})\} \right)$$

$$\leq \sum_{\substack{S \subset [n] \\ |S| = d-1}} P(\forall i \in [n] \setminus S \ X_i \in \text{span}(\{X_k \mid k \in S\})) \quad (26)$$

where the second inequality follows from the union bound. We now bound each of the terms of the sum. Fix $S = \{i_1, \ldots, i_{d-1}\} \subset [n]$ with $|S| = d - 1$. Let $Z_S = n((X_{i_j})_{j=1}^{d-1})$ be a non-zero vector orthogonal to $\text{span}(\{X_k \mid k \in S\})$. Such a vector must exist since $\dim(\text{span}(\{X_k \mid k \in S\})) < d$; see Lemma 8 below for an explicit construction of the function $n$. Denote by $P_{Z_S}$ the distribution of $Z_S$ and $P_{(X_i)_{i \in [n] \setminus S}} = \prod_{i=1}^{n-d-1} P_X$ the distribution of $(X_i)_{i \in [n] \setminus S}$, where $P_X$ is the distribution of $X$. Note that since $Z_S$ is a function of $(X_{i_j})_{j=1}^d$ only, it is independent of $(X_i)_{i \in [n] \setminus S}$. In particular, the joint distribution of $(Z_S, (X_i)_{i \in [n] \setminus S})$ is given by the product $P_{Z_S} \times P_{(X_i)_{i \in [n] \setminus S}}$. Now if $X_i \in \text{span}(\{X_k \mid k \in S\})$, then by definition of $Z_S$, $\langle Z_S, X_i \rangle = 0$. Therefore

$$P(\forall i \in [n] \setminus S \ X_i \in \text{span}(\{X_k \mid k \in S\})) \leq P(\forall i \in [n] \setminus S \ \langle Z_S, X_i \rangle = 0)$$

$$= E\left[ \prod_{i \in [n] \setminus S} \mathbb{1}_{\{0\}}(\langle Z_S, X_i \rangle) \right]$$

$$= \int \left\{ \prod_{i \in [n] \setminus S} \mathbb{1}_{\{0\}}(\langle y_S, x_i \rangle) \right\} P_{Z_S}(dz_S) P_{(X_i)_{i \in [n] \setminus S}}(d(x_i)_{i \in [n] \setminus S})$$

$$= \int P_{Z_S}(dy_s) \left\{ \prod_{i \in [n] \setminus S} \int \mathbb{1}_{\{0\}}(\langle y_S, x_i \rangle) P_X(dx_i) \right\}$$

$$= \int \left\{ \prod_{i \in [n] \setminus S} P(\langle z_S, X \rangle = 0) \right\} P_{Z_S}(dy_s)$$

$$= \int \left\{ \prod_{i \in [n] \setminus S} \rho_0(y_s) \right\} P_{Z_S}(dy_s)$$

$$\leq \rho^{n-d+1}, \quad (27)$$

where in the third line we used the independence of $Z_S$ and $(X_i)_{i \in [n] \setminus S}$, in the fourth we used the independence of the $(X_i)_{i \in \setminus S}$, in the sixth we used the definition of $\rho_0$ in 5, and in the last line we used the fact that $z_S \neq 0$ and the definition of $\rho$ in Lemma 1. Combining the inequalities (26) and (27) yields the result. $\qquad \square$

**Lemma 8.** *Let $m \in \{1, \ldots, d-1\}$ and let $(x_j)_{j=1}^m$ be a sequence of points in $\mathbb{R}^d$. Denote by $A \in \mathbb{R}^{m \times d}$ the matrix whose $j$-th row is $x_j$ and let $A^+$ be its pseudo-inverse. Let $(b_i)_{i=1}^d$ be an ordered basis of $\mathbb{R}^d$, and define*

$$k := \min\{i \in [n] \mid (I - A^+ A)b_i \neq 0\}$$
$$n((x_j)_{j=1}^m) := (I - A^+ A)b_k$$

*Then $n((x_j)_{j=1}^m)$ is non-zero and is orthogonal to $\operatorname{span}(\{x_j \mid j \in [m]\})$.*

*Proof.* We start by showing that $k$ is well defined. First note that $I - A^+ A$ is the orthogonal projector onto the kernel of $A$, which is non-trivial since $\dim(\ker(A)) = d - \dim(\operatorname{Im}(A)) \geq d - m \geq 1$. Now we claim that there exists an $i \in [d]$ such that $(I - A^+ A)b_i \neq 0$. Suppose not, then for any $w \in \mathbb{R}^d$, we have $(I - A^+ A)w = (I - A^+ A)(\sum_{i=1}^d c_i b_i) = \sum_{i=1}^d c_i(I - A^+ A)b_i = 0$, implying that $I - A^+ A = 0$, but this contradicts the non-triviality of $\ker(A)$. This proves that $k$ is well-defined, which in turn proves that $n((x_j)_{j=1}^m) \neq 0$. It remains to prove the orthogonality claim. Let $v \in \operatorname{span}(\{x_j \mid j \in [m]\})$. Then there exists coefficients $c \in \mathbb{R}^m$ such that $v = A^T c$. Therefore

$$\langle v, n((x_j)_{j=1}^m) \rangle = \langle A^T c, n((x_j)_{j=1}^m) \rangle = c^T A(I - AA^+)b_k = 0,$$

where the last equality holds since $(I - AA^+)b_k \in \ker(A)$. $\qquad\square$

# D  Missing proofs for Theorem 1

This section contains the proofs of Lemma 2 and Corollary 1 we used in the proof of Theorem 1.

## D.1  Proof of Lemma 2

We compute the expectation:

$$
\begin{aligned}
\mathrm{E}\Big[\|\nabla R_{p,n}(w_p^*)\|_{H_p^{-1}}^2\Big] &= \mathrm{E}\Big[\|n^{-1}\nabla \ell_p(\langle w_p^*, X_i\rangle - Y_i)\|_{H_p^{-1}}^2\Big] \\
&= n^{-2}\sum_{i=1}^n \mathrm{E}\Big[\|\nabla \ell_p(\langle w_p^*, X_i\rangle - Y_i)\|_{H_p^{-1}}^2\Big] \\
&\quad + 2n^{-2}\sum_{i=1}^n\sum_{j=1}^{i-1}\langle \mathrm{E}[\nabla \ell_p(\langle w_p^*, X_i\rangle - Y_i)], \mathrm{E}[\nabla \ell_p(\langle w_p^*, X_j\rangle - Y_j)]\rangle_{H_p^{-1}} \\
&= n^{-1}\mathrm{E}\Big[\|\nabla \ell_p(\langle w_p^*, X\rangle - Y)\|_{H_p^{-1}}^2\Big]
\end{aligned}
$$

where in the second line we expanded the inner product of the sums into its $n^2$ terms, used linearity of expectation, and used the independence of the samples to take the expectation inside the inner product. In the last line, we used the fact that the samples are identically distributed to simplify the first term. For the second term, we used the fact that the expectation of the gradient of the loss at the risk minimizer vanishes. Applying Markov's inequality finishes the proof. $\qquad\square$

## D.2  Proof of Corollary 1

We have

$$
\begin{aligned}
\|w - w_2^*\|_{H_{2,n}}^2 &= (w - w_2^*)^T H_{2,n}(w - w_2^*) \\
&= \frac{1}{n}\sum_{i=1}^n (w - w_2^*)^T \nabla^2 \ell_p(\langle w_2^*, X_i\rangle - Y_i)(w - w_2^*) \\
&= \frac{1}{n}\sum_{i=1}^n \langle w - w_2^*, X_i\rangle^2.
\end{aligned}
$$

Now by assumption, the components of the vector $X$ have finite fourth moment so that applying Proposition 1 and using the condition on $n$ yields the result. $\qquad\square$

# E Detailed proof of Theorem 4

This section contains the proof of Lemma 3 as well as that of Theorem 4.

## E.1 Proof of Lemma 3

By Lemma 2.5 in [AKPS22], we have for all $t, s \in \mathbb{R}$

$$\ell_p(t) - \ell_p(s) - \ell'_p(s)(t - s) \geq \frac{1}{8(p-1)} \ell''_p(s)(t - s)^2.$$

Recall that by the chain rule

$$\nabla \ell_p(\langle w, X \rangle - Y) = \ell'_p(\langle w, X \rangle - Y)X \qquad \nabla^2 \ell_p(\langle w, X \rangle - Y) = \ell''_p(\langle w, X \rangle - Y)XX^T.$$

Replacing $t$ and $s$ by $\langle w, X_i \rangle - Y_i$ and $\langle w_p^*, X_i \rangle - Y_i$ respectively, and using the formulas for the gradient and Hessian we arrive at

$$\ell_p(\langle w, X_i \rangle - Y_i) - \ell_p(\langle w_p^*, X_i \rangle - Y_i) \geq \frac{1}{8(p-1)}(w - w_p^*)^T \nabla^2 \ell_p(\langle w_p^*, X_i \rangle - Y_i)(w - w_p^*)$$
$$+ \langle \nabla \ell_p(\langle w_p^*, X_i \rangle - Y_i), w - w_p^* \rangle$$

Averaging over $i \in [n]$ yields the first inequality. The proof of the second inequality proceeds in the same way and uses instead the upper bound of Lemma 2.5 in [AKPS22]. We omit it here. $\qquad\square$

## E.2 Proof of Theorem 4

We proceed similarly to the proof of Theorem 1. By definition of the empirical risk minimizer, we have the upper bound

$$R_{p,n}(\hat{w}_p) - R_{p,n}(w_p^*) \leq 0. \tag{28}$$

Using (10) from Lemma 3 and the Cauchy-Schwarz inequality, we obtain the pointwise lower bound

$$R_{p,n}(\hat{w}_p) - R_{p,n}(w_p^*) \geq -\|\nabla R_{p,n}(w_p^*)\|_{H_p^{-1}} \|\hat{w}_p - w_p^*\|_{H_p} + \frac{1}{8(p-1)} \|\hat{w}_p - w_p^*\|_{H_{p,n}}^2. \tag{29}$$

Using Lemma 2 we have that, with probability at least $1 - \delta/2$,

$$\|\nabla R_{p,n}(w_p^*)\|_{H_p^{-1}} \leq \sqrt{2 \operatorname{E}\left[\|\nabla \ell_p(\langle w_p^*, X \rangle - Y)\|_{H_p^{-1}}^2\right]/(n\delta)}. \tag{30}$$

It remains to control $\|\hat{w}_p - w_p^*\|_{H_{p,n}}^2$ from below. Define the random vector

$$Z = |\langle w_p^*, X \rangle - Y|^{(p-2)/2} X$$

Then, for any $w \in \mathbb{R}^d$, we have

$$\|w - w_p^*\|_{H_{p,n}}^2 = (w - w_p^*)^T H_{p,n}(w - w_p^*)$$
$$= \frac{1}{n} \sum_{i=1}^n (w - w_p^*)^T \nabla^2 \ell_p(\langle w_p^*, X_i \rangle - Y_i)(w - w_p^*)$$
$$= \frac{1}{n} \sum_{i=1}^n \langle w - w_p^*, Z_i \rangle^2$$

By assumption, the components of the random vector $Z$ have finite fourth moment. Applying Proposition 1, and using the condition on $n$ assumed in the statement of Theorem 4, we get that, with probability at least $1 - \delta/2$, for all $w \in \mathbb{R}^d$,

$$\|w - w_p^*\|_{H_{p,n}}^2 \geq \frac{1}{2} \|w - w_p^*\|_{H_p}^2. \tag{31}$$

Combining (30) and (31) with (29) gives that with probability at least $1 - \delta$,

$$R_{p,n}(\hat{w}_p) - R_{p,n}(w_p^*) \geq -\sqrt{2 \operatorname{E}\Big[\|\nabla\ell_p(\langle w_p^*, X\rangle - Y)\|_{H_p^{-1}}^2\Big]/(n\delta)} \, \|\hat{w}_p - w_p^*\|_{H_p}$$

$$+ \frac{1}{16(p-1)}\|\hat{w}_p - w_p^*\|_{H_p}^2. \quad (32)$$

Further combining (32) with (28) and rearranging yields that with probability at least $1 - \delta$

$$\|\hat{w}_p - w_p^*\|_{H_p}^2 \leq \frac{512 p^2 \operatorname{E}\Big[\|\nabla\ell_p(\langle w_p^*, X\rangle - Y)\|_{H_p^{-1}}^2\Big]}{n\delta} \quad (33)$$

The last step is to bound the excess risk of the empirical risk minimizer using the bound (33) and (11) from Lemma 3. For that, we control the $L^p$ norm term in (11) as follows

$$p^p\|\hat{w}_p - w_p^*\|_{L^p}^p = \left(p^2 \frac{\|\hat{w}_p - w_p^*\|_{L^p}^2}{\|\hat{w}_p - w_p^*\|_{H_p}^2}\|\hat{w}_p - w_p^*\|_{H_p}^2\right)^{p/2}$$

$$\leq \left(p^2 \sup_{w \in \mathbb{R}^d \setminus \{0\}}\left\{\frac{\|w\|_{L^p}^2}{\|w\|_{H_p}^2}\right\}\|\hat{w}_p - w_p^*\|_{H_p}^2\right)^{p/2}$$

$$= \left(p^2 c_p^2\|\hat{w}_p - w_p^*\|_{H_p}^2\right)^{p/2}. \quad (34)$$

Combining (33), (11), and (34) yields the result. $\qquad\square$

# F    Detailed proof of Theorem 5

This section contains the proof of Lemma 4 as well as that of Theorem 5.

## F.1    Proof of Lemma 4

Both inequalities follow from Lemma 4.5 in Adil, Kyng, Peng, and Sachdeva [AKPS19]. (12) follows from a straightforward calculation using the lower bound of Lemma 4.5 in Adil, Kyng, Peng, and Sachdeva [AKPS19]; we omit it here. The upper bound requires a bit more work. We have by the quoted Lemma

$$\ell_p(t) - \ell_p(s) - \ell_p'(s)(t - s) \leq \frac{4}{p(p-1)}\gamma_p(|s|, |t - s|).$$

Now assume that $|s| > 0$. If $|t - s| \leq |s|$, we have

$$\gamma_p(|s|, |t - s|) = \frac{p}{2}|s|^{p-2}(t - s)^2 \leq |s|^{p-2}(t - s)^2 = \ell_p''(s)(t - s)^2.$$

Otherwise, if $|t - s| > |s|$, then we have

$$\gamma_p(|s|, |t - s|) = |t - s|^p - (1 - p/2)|s|^p \leq (t - s)^2|t - s|^{p-2} \leq |s|^{p-2}(t - s)^2 = \ell_p''(s)(t - s)^2.$$

Therefore in both cases we have $\gamma_p(|s|, |t - s|) \leq \ell_p''(s)(t - s)^2$ as long as $|s| > 0$. Replacing $t$ and $s$ by $\langle w, X\rangle - Y$ and $\langle w_p^*, X\rangle - Y$ respectively we get, on the event that $\langle w_p^*, X\rangle - Y \neq 0$

$$\ell_p(\langle w, X\rangle - Y) - \ell_p(\langle w_p^*, X\rangle - Y) - \langle \nabla\ell_p(\langle w_p^*, X\rangle - Y), w - w_p^*\rangle \leq \frac{4}{p(p-1)}\|w - w_p^*\|_{\nabla^2\ell_p(\langle w_p^*, X\rangle - Y)}$$

Recalling that by assumption $\operatorname{P}\big(\langle w_p^*, X\rangle - Y \neq 0\big) = 1$, taking expectation of both sides, and bounding $1/p \leq 1$ finishes the proof of (13). $\qquad\square$

### F.2 Proof of Theorem 5

We follow the same proof strategy as the one used in the proofs of Theorems 1 and 4. By definition of the empirical risk minimizer, we have

$$R_{p,n}(\hat{w}_p) - R_{p,n}(w_p^*) \leq 0. \tag{35}$$

Using (12) from Lemma 4 and the Cauchy-Schwarz inequality, we have the lower bound

$$R_{p,n}(\hat{w}_p) - R_{p,n}(w_p^*) \geq -\|\nabla R_{p,n}(w_p^*)\|_{H_p^{-1}} \|\hat{w}_p - w_p^*\|_{H_p}$$
$$+ \frac{1}{4p^2} \frac{1}{n} \sum_{i=1}^{n} \gamma_p\big(|\langle w_p^*, X_i \rangle - Y_i|, |\langle w - w_p^*, X_i \rangle|\big) \tag{36}$$

Using Lemma 2 we have that, with probability at least $1 - \delta/2$,

$$\|\nabla R_{p,n}(w_p^*)\|_{H_p^{-1}} \leq \sqrt{2 \, \mathrm{E}\Big[\|\nabla \ell_p(\langle w_p^*, X \rangle - Y)\|_{H_p^{-1}}^2\Big]/(n\delta)}. \tag{37}$$

On the other hand, by Proposition 2, we have with probability $1 - \delta/2$, for all $w \in \mathbb{R}^d$,

$$\frac{1}{n} \sum_{i=1}^{n} \gamma_p\big(|\langle w_p^*, X_i \rangle - Y_i|, |\langle w - w_p^*, X_i \rangle|\big) \geq \frac{p}{8} \min\Big\{\|w - w_p^*\|_{H_p}^2, \varepsilon^{2-p} \|w - w_p^*\|_{H_p}^p\Big\}, \tag{38}$$

where $\varepsilon$ is as defined in Proposition 2. We now consider two cases. If $\|\hat{w}_p - w_p^*\|_{H_p}^2 \leq \varepsilon^{2-p} \|\hat{w}_p - w_p^*\|_{H_p}^p$, then combining (35), (36), (37), and (38) gives

$$\|\hat{w}_p - w_p^*\|_{H_p}^2 \leq \frac{8192 \, \mathrm{E}\Big[\|\nabla \ell_p(\langle w_p^*, X \rangle - Y)\|_{H_p^{-1}}^2\Big]}{n\delta}. \tag{39}$$

Otherwise, $\|\hat{w}_p - w_p^*\|_{H_p}^2 > \varepsilon^{2-p} \|\hat{w}_p - w_p^*\|_{H_p}^p$, then again combining (35), (36), (37), and (38) gives

$$\|\hat{w}_p - w_p^*\|_{H_p}^2 \leq \left(\frac{8192 \, \mathrm{E}\Big[\|\nabla \ell_p(\langle w_p^*, X \rangle - Y)\|_{H_p^{-1}}^2\Big] \varepsilon^{2(p-2)}}{n\delta}\right)^{1/(p-1)} \tag{40}$$

In either case, we have, using (39) and (40), with probability at least $1 - \delta$,

$$\|\hat{w}_p - w_p^*\|_{H_p}^2 \leq \frac{8192 \, \mathrm{E}\Big[\|\nabla \ell_p(\langle w_p^*, X \rangle - Y)\|_{H_p^{-1}}^2\Big]}{n\delta}$$
$$+ \left(\frac{8192 \, \mathrm{E}\Big[\|\nabla \ell_p(\langle w_p^*, X \rangle - Y)\|_{H_p^{-1}}^2\Big] \varepsilon^{2(p-2)}}{n\delta}\right)^{1/(p-1)}.$$

Combining this last inequality with (13) from Lemma 4 finishes the proof. $\qquad\square$

