# OpenReview forum: "Optimal Excess Risk Bounds for Empirical Risk Minimization on $p$-Norm Linear Regression"
_NeurIPS.cc/2023/Conference — NeurIPS 2023 poster_

### Official Review · Reviewer_Ydss · 2023-07-06

**Soundness:** 3 good
**Presentation:** 3 good
**Contribution:** 3 good
**Rating:** 7
**Confidence:** 3

**Summary:**

The paper studies the excess risk of empirical risk minimization on the $p$-norm linear regression. The asymptotic bound of the excess risk of ERM for $p$ in ($1,\infty$) is known by prior work and more recent work had given high probability excess risk bounds for $p=2$ that match that asymptotic rate. This paper expands the result to $p$ in $(1,\infty) \setminus \{2\}$. It splits the results into three parts: It first provides bounds for the realizable case for $p \in$ (1,$\infty$). Then, a bound for the non-realizable case with $p$ $\in$ (2,$\infty$) where it requires mild assumptions on the moments that are natural extensions of the assumptions needed for the $p=2$ prior work’s result. For the non-realizable case with $p$ $\in$ (1,2), the paper requires some additional assumptions: a slightly stronger version of non-realizability, assuming that the inner product of the covariates $x$ with the optimal minimizer $w^*$ is almost surely not equal to the label, and the existence of the $2(2-p)$ negative moment of $y-\langle w^*,x \rangle$.

The proof technique draws inspiration from Oliveira 2016, Lecue and Mendelson 2016b, which by itself is not sufficient. However, when combined with a recent result of Adil et al. 2022, the approach becomes valid by allowing the use of the second-order Taylor expansion, despite the fact that it is not exact. For the case of $p$ $\in$ (1,2), a different approach is employed, utilizing an approximation for the $p$-th power function from Bubeck et al. 2018, along with the analysis of Adil et al. 2019a.


**Strengths:**

The paper provides the first high probability excess risk bounds of ERM for $p$-norm regression for $p \in (1,\infty)\setminus \{2\}$ that are consistent with the existing asymptotic bounds. This is a considerable result. The paper does a very good job in describing the problem’s motivation, the problem’s background and the new results.

**Weaknesses:**

As the authors have already described, the case $p \in (1,2)$ requires extra assumptions that it is unclear if they are necessary, leaving the goal of the paper for that regime only partially resolved.

Although the paper writing is good in presenting the problem and the results, the proof strategy of the main results is not really discussed until later sections of the paper.

As a minor comment on writing, it would be better to provide a definition of $\rho$ (or a reference to its definition) inside the theorem statements that use it. Similar comment for $\sigma_p^2$.


**Questions:**

What do the three regimes for $\rho$ represent in Theorem 3?

**Limitations:**

Addressed.

---

> ### Author Rebuttal · Authors · 2023-08-10
>
> We thank the reviewer for their positive evaluation and constructive feedback. We will make sure to add a definition of $\rho$ and $\sigma_p^2$ inside the statements of the theorems to make them more readable.
>
> ---
>
> ***”What do the three regimes represent in Theorem 3?”***
>
> They are an artifact of the particular probability bound we obtain in Theorem 3. Specifically, the three cases are needed to properly capture the dominating term in the sample complexity. This is due to the relative growth of the binomial coefficients compared to the exponential with base $\rho$, which varies as $\rho$ varies from 0 to 1.
>
> ---
> We would be happy to clarify any concerns or answer any questions that may come up during the discussion period.

---

### Official Review · Reviewer_YnMx · 2023-07-24

**Soundness:** 4 excellent
**Presentation:** 3 good
**Contribution:** 2 fair
**Rating:** 6
**Confidence:** 3

**Summary:**

This paper studies the performance of empirical risk minimization (ERM) for $\ell_p$-norm linear regression, in both the realizable and non-realizable (say, noisy measurement) settings. While the $p=2$ case has been studied extensively in the literature, not as much is known for the general $p \in (1,2) \cup (2,\infty)$ cases. For the realizable case, the author(s) improve over prior analyses and give a $O(d)$ sample complexity bound for finding the regression coefficients with high probability. For the non-realizable case, under moment assumptions, this paper shows high probability excess risk bounds.

Note: this is an emergency review, and so I did not check the math carefully.

**Strengths:**

The paper is well-written and clear. The author(s) motivate the $\ell_p$ linear regression problem well, and this paper gives the first non-asymptotic excess risk bounds for ERM in this setting. The proof ideas are also explained pretty clearly.

**Weaknesses:**

My main concern/confusion is the message of the paper. The author(s) seem to sell the paper as having a high probability (excess) risk bound, yet the bound has a $1/\delta$ dependence on the failure probability $\delta$ (from using Markov, and might actually be tight) instead of logarithmic or root-logarithmic (sub-Gaussian) performance. ERM is very susceptible to heavy-tailed noise, even in the simple special case of mean estimation, so I'm a bit confused why we're studying the performance of ERM here, instead of designing better estimators that work under such mild moment assumptions. From my readthrough, it also appears that the paper is really bounding some expectation and using Markov to get the high probability bound. My question then, is, why is the paper phrased as a high probability result instead of an expectation/constant probability result, which I can imagine ERM to actually be decent at?

**Questions:**

Beyond the motivation question I have above:

- The results are being compared to the asymptotic guarantees of ERM, but is there any evidence that ERM's asymptotic performance is statistically optimal (even when compared with other estimators)? In the special case of mean estimation, for example, we can actually write down non-asymptotic lower bounds on the estimation error in the Gaussian cases, which match the central limit theorem.
- Can the author(s) further elaborate on the extra moment assumption $\mathbb{E}[|Y - \langle w_p^*,X\rangle|^{2(p-2)} X_j^4] < \infty$ in Theorem 4? Line 168 states that it's a "natural extension", but really doesn't give interpretation to this assumption, in particular, the interaction between the response and the covariate coordinates.

Another misc comment: The beginning of Section 2.2 makes it sound like Theorem 1 is an old result, but after the theorem, the author(s) state that there's novelty at least in the proof presentation. Probably worth writing that more explicitly before the theorem statement.

---

> ### Author Rebuttal · Authors · 2023-08-10
>
> We thank the reviewer for their valuable feedback. Please find our detailed response below.
>
> ---
>
> ***”My main concern/confusion is the message of the paper.”***
>
> The goal and achievement of the paper is the study of the performance of ERM non-asymptotically and under weak moment assumptions for the $p$-norm regression problem.
> We motivate the ERM setting in the next paragraph.
>
> ***”I'm a bit confused why we're studying the performance of ERM here, instead of designing better estimators that work under such mild moment assumptions.”***
>
> We list a subset of the reasons why one would like to study ERM before trying to develop more sophisticated estimators with the optimal subgaussian performance under heavy tails.
>
> - Firstly, ERM is the most widely used method in practice, not the least because of its computational tractability, and so it is interesting to understand its performance and its failure modes.
>
> - Secondly, we decided to present our results under the weakest moment assumptions we could work with, which is why we incur the suboptimal $1/\delta$ dependence on the confidence parameter. Note however that under stronger assumptions, e.g. subgaussianity of $X$ and $Y$, one may easily show root-logarithmic dependence on $\delta$ using our proof techniques: it is enough to exploit the subgaussianity of $X$ and $Y$ to prove tighter concentration in Lemma 2. In such cases, which might occur in practical problems, ERM is likely to be competitive with the yet to be discovered, more sophisticated, "subgaussian" estimators.
>
> - Thirdly, just like the proof techniques used for the study of ERM for the case p=2 were very useful later in proving guarantees for more sophisticated subgaussian estimators, we believe that our proof techniques will serve as the basic machinery to study the performance of yet to be proposed estimators with subgaussian performance. In this sense, we view our work as a stepping stone towards the end goal the reviewer has outlined.
>
> ---
>
> ***”From my readthrough, it also appears that the paper is really bounding some expectation and using Markov to get the high probability bound. My question then, is, why is the paper phrased as a high probability result instead of an expectation/constant probability result, which I can imagine ERM to actually be decent at ?”***
>
> In short, we do not have a bound on the expectation of the excess risk, and deriving such a bound requires non-standard assumptions. In more detail, we do not bound the expectation of the excess risk then use Markov's inequality, but rather combine two high-probability bounds: one of them is indeed coming from a bound on an expectation (Lemma 2), the other however is a uniform probability estimate (Proposition 1), and is in fact weaker than a bound on the expectation. Roughly speaking, this is because a bound on the expectation requires controlling the lower tail of the smallest eigenvalue of the empirical covariance matrix at all levels, whereas Proposition 1 provides a non-trivial bound only at a fixed level. Getting control over all levels requires non-standard assumptions (in particular, a quantitative version of the small-ball condition), please see (Mourtada, 2022) for such an approach and a thorough discussion of this issue.
>
> ---
>
> ***”The results are being compared to the asymptotic guarantees of ERM, but is there any evidence that ERM's asymptotic performance is statistically optimal (even when compared with other estimators)?”***
>
> Our goal and emphasis is to study the performance of ERM, so it is natural for us to compare our non-asymptotic guarantees with the asymptotic ones. The optimality of our bounds that we claim is in the sense that the leading term of our non-asymptotic bounds matches the asymptotically exact expression for the excess risk of ERM given by equation (1).
>
> Concerning the statistical optimality of ERM, if the reviewer is referring to minimax optimality over a given class of distributions, please see (Mourtada, 2022) for results for the case $p = 2$, where it is proven that, under mild regularity conditions, ERM is minimax optimal for well-specified models, and is asymptotically minimax optimal for misspecified ones. While we agree that it would be nice to extend such results to the cases $p \neq 2$, this is outside the scope of our paper, and is an interesting future work.
>
> ---
>
> ***”Can the author(s) further elaborate on the extra moment assumption [...]?”***
>
> It is an extension in the sense that when $p = 2$, this assumption reduces to the fourth moment condition of Theorem 1. We are afraid there is no straightforward "intuitive" interpretation of this assumption in terms of the interaction between the response and covariates. Nevertheless we qualified the assumption as natural in that it arises directly from the proof and in a completely analogous way to the proof of the case $p=2$.
>
> ---
> We would be happy to clarify any concerns or answer any questions that may come up during the discussion period.

---

> > ### Comment · Reviewer_YnMx · 2023-08-10
> >
> > I thank the authors for the response. I appreciated in particular the clarification on why the result is phrased as a $1/\delta$-dependence result as opposed to an expectation result.
> >
> > I have follow-up questions and comments:
> >
> > 1. Could the authors point me to how the study of ERM in $\ell_2$ regression led to sub-Gaussian estimators in that setting?
> >
> > 2. Asymptotic vs non-asymptotic guarantees: my question is actually, is the asymptotic rate of ERM the minimax rate, even if it is the minimax rate attained by another estimator? Ignoring the $\delta$ dependence, are there lower bounds on this minimax rate (for general $p$)?
> >
> > 3. It seems very unsatisfactory that there is an uninterpretable assumption in a main result. Can the author(s) say anything more beyond "this is how we get the proof to go through"?

---

> > > ### Author Response · Authors · 2023-08-10
> > >
> > > We thank the reviewer for following up on our rebuttal. Please find our response to the additional questions below.
> > >
> > > ---
> > >
> > > ***"1- Could the authors point me to how the study of ERM in $\ell_2$ regression led to sub-Gaussian estimators in that setting?"***
> > >
> > > Our claim was that the proof techniques used to study the performance of ERM in the case $p=2$ were useful later in proving the subgaussian performance of newly proposed estimators.
> > >
> > > Indeed, the decomposition of the excess $\ell_{2}$ loss in terms of quadratic and multiplier processes (see, e.g., equation (2.1) in Lecue and Mendelson, 2016) was first proposed in the context of the study of ERM (Lecue and Mendelson, 2016 and references therein), and was reused, among others, in the analysis of the first subgaussian estimator proposed by (Lugosi and Mendelson, 2016) (see also Lugosi and Mendelson, 2019) and subsequently for the subgaussian estimator proposed by (Lecué and Lerasle, 2020).
> > >
> > > Furthermore, this decomposition, coupled with the realization that the quadratic process is lower bounded with high probability even in the presence of heavy tails (Koltchinskii and Mendelson, 2015; Oliveira, 2016), helped isolate the weak concentration of the multiplier process in ERM (equivalent of Lemma 2 in our paper) as the reason for its suboptimality. Newly proposed algorithms focused on better estimating this component of the excess loss, thereby achieving the desired subgaussian performance.
> > >
> > > In summary, the study of the performance of ERM contributed to the development of subgaussian estimators in multiple ways.
> > >
> > > - It motivated their development since ERM was found to suffer a suboptimal dependence on the confidence parameter $\delta$ in the case of heavy-tails (see, e.g., the discussion in Sec 1.1. in Lugosi and Mendelson, 2016).
> > > - It helped isolate the reason for this suboptimality (the weak concentration of the multiplier process discussed above).
> > > - It provided the basic technical machinery to prove the subgaussian performance of newly proposed estimators (among others, the excess loss decomposition, and the control of the quadratic process from below under heavy tails).
> > >
> > > ---
> > >
> > > ***"2- Asymptotic vs non-asymptotic guarantees: my question is actually, is the asymptotic rate of ERM the minimax rate, even if it is the minimax rate attained by another estimator? Ignoring the $\delta$ dependence, are there lower bounds on this minimax rate (for general $p$)?"***
> > >
> > > To the best of our knowledge, the minimax rate for $\ell_{p}$ norm regression is unknown for $p \neq 2$, and we are not aware of any lower bounds on it. We are therefore not in position to say whether the asymptotic rate of ERM is minimax or not for general $p$. As discussed above, the case $p=2$ was only very recently settled by Mourtada, (2022), and to the best of our knowledge, it remains an interesting open problem for other values of $p \in (1, \infty)$.
> > >
> > > ---
> > > **References**
> > >
> > > Guillaume Lecué, and Shahar Mendelson. "Performance of empirical risk minimization in linear aggregation." (2016): 1520-1534.
> > >
> > > Gábor Lugosi and Shahar Mendelson. "Risk minimization by median-of-means tournaments." Journal of the European Mathematical Society 22.3 (2019): 925-965.
> > >
> > > Gábor Lugosi and Shahar Mendelson. "Mean estimation and regression under heavy-tailed distributions: A survey." Foundations of Computational Mathematics 19.5 (2019): 1145-1190.
> > >
> > > Guillaume Lecué and Matthieu Lerasle. "Robust machine learning by median-of-means: theory and practice." (2020): 906-931.
> > >
> > > Vladimir Koltchinskii and Shahar Mendelson. "Bounding the smallest singular value of a random matrix without concentration." International Mathematics Research Notices 2015.23 (2015): 12991-13008.
> > >
> > > Oliveira, Roberto Imbuzeiro. "The lower tail of random quadratic forms with applications to ordinary least squares." Probability Theory and Related Fields 166 (2016): 1175-1194.
> > >
> > > Jaouad Mourtada. "Exact minimax risk for linear least squares, and the lower tail of sample covariance matrices." The Annals of Statistics 50.4 (2022): 2157-2178.

---

> > > > ### Author Response · Authors · 2023-08-11
> > > >
> > > > ***"3- It seems very unsatisfactory that there is an uninterpretable assumption in a main result. Can the author(s) say anything more beyond "this is how we get the proof to go through"?"***
> > > >
> > > > The moment assumption $\mathbb{E}[|Y-\langle w_p^*, X\rangle|^{2(p-2)} X_j^4] <\infty$ is a condition on the joint distribution of the random variables $Y$ and $X$. We consider below cases of increasing generality that make it more interpretable.
> > > >
> > > >
> > > > **Well-specified case:** Assume that $Y = \langle w^*, X\rangle + \varepsilon$ for noise $\varepsilon$ independent of $X$ and whose distribution is symmetric around $0$. One may show under mild regularity conditions (see Lai and Lee, 2005) that $w_p^* = w^*$ for all $p \in (1, \infty)$. In this case, our assumption decouples nicely as
> > > >
> > > > &nbsp;&nbsp;&nbsp;&nbsp;&nbsp;&nbsp;&nbsp;&nbsp;&nbsp;&nbsp;&nbsp;&nbsp;&nbsp;&nbsp;&nbsp;$\mathbb{E}[|Y-\langle w_p^*, X\rangle|^{2(p-2)} X_j^4] <\infty \iff \mathbb{E}[X_j^{4}] <\infty \text{ and } \mathbb{E}[\varepsilon^{2(p-2)}]<\infty. $
> > > >
> > > > The first assumption is simply the standard fourth moment assumption on the covariates, which is used in the case $p=2$ (Oliveira, 2016). The second is an assumption on the noise $\varepsilon$. For $p \in (2, \infty)$, this assumption requires that the distribution of $\varepsilon$ is not too heavy tailed by requiring the existence of $2(p-2)$ moments. For $p \in (1, 2)$, this assumption requires that the distribution of $\varepsilon$ is not too concentrated around $0$ so that the negative $2(2-p)$ moment of $\varepsilon$ exists.
> > > >
> > > > **General case:** In this case, the assumption does not decouple as nicely as in the well-specified case. However, by Hölder's inequality, for any $q, r \in [1, \infty]$ satisfying $1/q + 1/r = 1$, we have the bound
> > > >
> > > > &nbsp;&nbsp;&nbsp;&nbsp;&nbsp;&nbsp;&nbsp;&nbsp;&nbsp;&nbsp;&nbsp;&nbsp;&nbsp;&nbsp;&nbsp; $\mathbb{E}[|Y-\langle w_p^*, X\rangle|^{2(p-2)} X_j^4] \leq \mathbb{E}[\varepsilon^{2(p-2) q}]^{1/q} \mathbb{E}[X_j^{4r}]^{1/r},$
> > > >
> > > > where $\varepsilon := Y - \langle w_{p}^* , X\rangle$. Therefore, our assumption is **implied** by the finiteness of the raw moments $\mathbb{E}[\varepsilon^{2(p-2)q}]$ and $\mathbb{E}[X_j^{4r}]$, for any valid choices of $q$ and $r$. These raw moment assumptions can be given similar interpretations to the moment assumptions discussed in the well-specified case. Furthermore, the choice of $q$ and $r$ introduces a tradeoff: a larger $q$ constrains the distribution of $\varepsilon$ more severely, while a larger $r$ constrains the distribution of the covariates $X$ more severely.
> > > >
> > > > If the reviewer finds the above discussion useful, we would be happy to include it in the paper.
> > > >
> > > > ---
> > > >
> > > > We would be happy to answer any further questions that may come up.
> > > >
> > > > ---
> > > >
> > > > **References**
> > > >
> > > > Oliveira, Roberto Imbuzeiro. "The lower tail of random quadratic forms with applications to ordinary least squares." Probability Theory and Related Fields 166 (2016): 1175-1194.
> > > >
> > > > P. Y. Lai and Stephen M. S. Lee. "An overview of asymptotic properties of Lp regression under general classes of error distributions." Journal of the American Statistical Association 100.470 (2005): 446-458.

---

> > > > > ### Comment · Reviewer_YnMx · 2023-08-11
> > > > >
> > > > > I thank the authors for further engaging. I think this discussion has been important for contextualizing the work, and should be included in the paper (even if some of it is in the appendix, but pointed to in the main paper). I'm happy to raise my score to 6.
> > > > >
> > > > > I have some final comments, including a single question:
> > > > >
> > > > > 1. The interpretation of the assumption was exactly the kind of intuition I was hoping the authors to give to the reader. The intuition is much cleaner than I thought I would be, which is great, and is very helpful for a reader who just want to understand the main point of the paper without taking a deep dive. This, in my opinion, is the most important to include in the paper.
> > > > >
> > > > > 2. I should've been a bit clearer re: minimax rate. I was curious about the minimax rate of the high-probability error, up to constants (for now), whether that's known? Mourtada (2022) seems to be addressing the mean squared error instead, and the point was to characterize the minimax rate without losing any constants.
> > > > >
> > > > > 3. I think it's also good to include the "study of ERM can help find sub-Gaussian estimators" motivation, in order to sell this work as a high probability result. I would omit the point about "realizing ERM is sub-optimal" as a motivation though, since that much is clear to the community at this point in time, but it's good to point it out in the paper anyway for contextualizing the result. It'd also really strengthen the paper if the authors could include some comments about which part of their analysis might be useful for the pursuit of sub-Gaussian estimators.

---

> > > > > > ### Author Response · Authors · 2023-08-12
> > > > > >
> > > > > > We thank the reviewer for engaging with our answers and providing valuable feedback and suggestions. We will include the ideas we have discussed in our exchange in the paper, and acknowledge the contribution of the reviewer in the appropriate section. We address your question below.
> > > > > >
> > > > > > ---
> > > > > >
> > > > > > ***"I was curious about the minimax rate of the high-probability error, up to constants (for now), whether that's known?"***
> > > > > >
> > > > > > One of the few results we know of in the direction you are describing is Theorem A' in (Lecué and Mendelson, 2013), but unfortunately, it is specific to the case $p=2$. To the best of our knowledge, such results are not known for the cases $p \in (1, 2) \cup (2,\infty)$, and characterizing the minimax rate in this case (both in expectation and in high-probability) remains an open problem.
> > > > > >
> > > > > > ---
> > > > > >
> > > > > > **References**
> > > > > >
> > > > > > Guillaume Lecué and Shahar Mendelson. "Learning subgaussian classes: Upper and minimax bounds." arXiv preprint arXiv:1305.4825 (2013).

---

### Official Review · Reviewer_81uA · 2023-07-26

**Soundness:** 2 fair
**Presentation:** 2 fair
**Contribution:** 2 fair
**Rating:** 5
**Confidence:** 4

**Summary:**

This paper studies the empirical risk minimization of linear regression with the p-norm loss function, where p ∈ (1,∞).

Main contributions:
- The authors provide a high probability excess risk bound on the empirical risk minimizer, for p ∈ (1,∞), where for p ∈ [2,∞) only weak moment assumptions are needed, and for p ∈ (1, 2) there are assumptions that guarantee the existence of the hessian at the minimizer of the risk.
- The authors strengthen a previous bound of perfect recovery in the realizable case.

**Strengths:**

Some strengths are:
- This paper is the first work providing the high probability excess risk bounds for empirical risk minimization under the p-norm loss function with any p ∈ (1,∞).
- The authors strengthen a previous bound of perfect recovery in the realizable case.

**Weaknesses:**

Some weaknesses of the paper:
- The proof part is poorly written. For example, the statements of Corollary 1 and Proposition 1 don't make any sense (the inf of the left-hand side is less than the right-hand side?), where even for Proposition 1 the authors just want to cite an existing theorem in a previous paper (Theorem 3.1, Oliveira [2016]).
- There is a lack of the argument of the importance of this work. The authors discuss the benefits of using p-norm as the loss with p != 2, but why moving from asymptotic bound to finite sample bound is important for p != 2 is missing.

Typos:
- At line 232, second line of the equation misses index i on X and Y.

**Questions:**

NA

---

> ### Author Rebuttal · Authors · 2023-08-10
>
> We thank the reviewer for their valuable feedback. We kindly ask the reviewer to re-evaluate the paper in light of our response below.
>
> ---
>
> ***”The proof part is poorly written.”***
>
> We thank the reviewer for pointing us to the typos in Corollary 1 and Proposition 1; we fixed these in the supplementary version, but we understand that this must have been a source of confusion when reading the original submission. To be more explicit, the required changes are as follows.
>
> - For Proposition 1, it suffices to move the term $\||v\||_{L^{2}}^{2}$ from the RHS of the statement to the denominator of the LHS.
> - Similarly, for Corollary 1, the term $\||w - w_{2}^{*}\||^2_{H_{2}}$ on the RHS should be moved to the denominator of the LHS.
>
> The proofs of both statements remain unchanged. We invite the reviewer to consult the supplementary version for a correct and clearer version of both statements.
>
> Please note that we have attempted to formulate our proofs in an accessible and readable way, and have dedicated a lot of effort in motivating all the steps in the proofs, emphasizing where our proofs differ from previous ones, and where our insights were needed. If the reviewer has other specific comments about the proofs, we would be happy to address them.
>
>
> ***”There is a lack of the argument of the importance of this work. The authors discuss the benefits of using p-norm as the loss with p != 2, but why moving from asymptotic bound to finite sample bound is important for p != 2 is missing.”***
>
> We clarify this with the following sentence in the revised version of our paper.
>
> *Among other reasons, non-asymptotic bounds are useful in characterizing **when** the asymptotic regime kicks in, and under what assumptions it can be attained for moderately large $n$, see, e.g., (Ostrovski and Bach, 2021) for more on this argument.*
>
> ---
> We would be happy to clarify any concerns or answer any questions that may come up during the discussion period.

---

> ### Comment · Reviewer_81uA · 2023-08-20
>
> I thank the authors for the response. I'm willing to increase the score to 5.

---

### Official Review · Reviewer_KiJP · 2023-07-26

**Soundness:** 3 good
**Presentation:** 3 good
**Contribution:** 3 good
**Rating:** 6
**Confidence:** 2

**Summary:**

The paper considers linear regression with inputs in dimension d and with n data points. It consider the Lp loss |.|^p with a focus on p different from the classical value 2. The paper provides non-asymptotic risk excess bound for this Lp loss, the excess being between the optimal linear combination w^* and the one obtained by minimizing the empirical risk. The bounds are of different natures in the two cases where the problem is realizable, meaning that the output is exactly a linear function of the input, and where the problem is not realizable.

**Strengths:**

The paper is well written. The problem is theoretically interesting. The theoretical results are well-connected to the existing literature and their novelty compared to this literature is well motivated. The proofs appear to be innovative and to combine arguments from different branches of  the literature.

**Weaknesses:**

The content is technical and so the paper is not extremely easy to follow.
An option could be to take advantage of the appendix (with no space limit) to add more slow-paced pedagogical content.

**Questions:**

I do not have specific questions.

**Limitations:**

I do not see potential negative societal impact.
The limitations are discussed adequately

---

> ### Author Rebuttal · Authors · 2023-08-10
>
> We thank the reviewer for their positive evaluation. We will be happy to answer any questions or comments that may arise during the discussion period.

---

### Official Review · Reviewer_1SnJ · 2023-07-26

**Soundness:** 3 good
**Presentation:** 3 good
**Contribution:** 3 good
**Rating:** 7
**Confidence:** 4

**Summary:**

The paper studies excess risk bounds for empirical risk minimization for linear regression under $\ell_p$-norm. It extends existing work $p=2$ to general $p\in (1,\infty)$.

Sampling complexity bounds are given for recovering the true parameter in the realizable case.

Additive error excess risk bounds are given (under mild assumptions on $O(p)^{th}$ moments) in the non-realizable case

I generally like the paper but some issues need to be discussed/resolved, based on which my rating could move either way.

**Strengths:**

* natural extension to $p$-norm regression
* non-asymptotic bounds take similar forms as the asymptotic results in the p=2 case (up to dependencies on constants and $p$)
* rigorous theory
* good high-level discussion on the difficulties of generalizing previous proofs to general $p$ and how they are handled
* good overview over existing work

**Weaknesses:**

* "Optimality", claimed already in the title is not supported by any (existing or new) lower bounds
* some assumptions seem to 'eat' terms, that depend on the dimension d. There is some, but only very limited discussion on this
* no result for the second most important case $p=1$
* in a few places very 'mathematical' statements without explaining the semantic or intuition behind
* bounds depend on Hessian forms of $\ell_p$ loss functions, which fit _perfectly well_ to the quadratic function in the $p=2$ case but do not capture the loss as closely for other $p$, even though the bounds look similar
* discussion on (third paragraph, ll 23-32) motivating the $\ell_p$ loss regression problem could be improved and seems vague wrt the properties of the model. I suggest rewriting this, stating that it is the standard linear model with $p$-generalized Gaussian error distribution that has certain statistical properties parameterized by $p$. See https://arxiv.org/pdf/2203.13568.pdf and references therein.

**Questions:**

1. Optimality is a strong claim. In what sense can your bounds be considered _optimal_ ? I see no lower bounds at all !?!

2. Theorem 3 seems (at first glance) even below known lower bounds. For instance if the generating distribution realizes standard basis vectors with equal probability and the target is not contained in any subspace spanned by a proper subset, it is known that $\Omega(d \log d)$ is necessary to recover the exact target (follows simply from the coupon collector's theorem). So, my question is, what is the role and the dependence of the $\rho$ parameter on the dimension?

3. Again Theorem 3: the three cases seem a bit arbitrary, why is it necessary to distinguish between them? is it only to make the upper bounds work in any case or are there stronger indications for actually separated complexities in the three cases?

4. Why are there no bounds on $p=1$, would there be a major complication (in your analysis or in general) in tackling this case?

5. What is a "negative moment"?

**Limitations:**

The very important case $p=1$ is not handled or discussed at all beyond the intro/motivation

---

> ### Author Rebuttal · Authors · 2023-08-10
>
> We thank the reviewer for their positive evaluation and detailed feedback. We address the reviewer’s concerns below.
>
> ---
>
> ***”Optimality is a strong claim. In what sense can your bounds be considered optimal?”***
>
> Our bounds are optimal in the sense that the leading term in them matches the asymptotically **exact** expression of the excess risk of ERM given by equation (1) as $n \to \infty$, as we described after the statements of Theorems 4 and 5 (lines 164-165 and 181-182). In other words, **any** bound on the excess risk that holds for large enough $n$ must dominate the first term in the RHS of equation (1); our bounds are optimal in that they exactly recover this term (up to constants that depend only on $p$).
>
> ---
>
> ***”What is the role and the dependence of the parameter $\rho$ on the dimension?”***
>
> The role of the parameter $\rho$ is to quantify the degree of "degeneracy'' of the distribution of the covariates (see, e.g., Mourtada 2022), i.e., how well can the support of the distribution of the covariates be approximated by some fixed hyperplane.
>
> In general, $\rho$ does not depend directly on the dimension. Indeed, on the one hand, and when the distribution of the covariates $X$ is absolutely continuous with respect to Lebesgue measure, $\rho = 0$, no matter how large the dimension is. On the other hand, $\rho$ can be arbitrarily close to 1, even in very low dimensional settings. Indeed, let $\varepsilon \in (0, 1)$, and consider one dimensional $X$ satisfying $X = 0$ with probability $1 - \varepsilon$, and $X = 1$ with probability $\varepsilon$, then $\rho = 1 - \varepsilon$.
>
> Finally, let us mention that this parameter and closely related ones have been used many times in previous work (Rudelson et al., 2015; Lecue and Mendelson, 2017; Mourtada 2022), and that from a technical perspective, as we have discussed in the paper (lines 129-139), it is a natural parameter to consider.
>
> ---
>
> ***”Why is it necessary to distinguish between the three cases in Theorem 3? Is it only to make the upper bounds work in any case or are there stronger indications for actually separated complexities in the three cases?”***
>
> It is only to make the upper bound on the failure probability work.
>
> ---
>
> ***”Some assumptions seem to "eat" terms that depend on the dimension d. There is some, but only very limited discussion on this.”***
>
> If the reviewer is referring to the constants arising in our bounds, please note that we discussed their dimension dependence right after the statements of Theorems 4 and 5 (lines 167-171 and 196-197), and how that affects the optimality of our bounds. As for the constants $\sigma^2_{p}$, they are powers of norm equivalence constants; one can show that they can be arbitrarily bad, even in a one dimensional setting, using an example similar to the one we used above to demonstrate the case $\rho=1-\varepsilon$. On the other hand, and to the best of our knowledge, no dimension dependent lower bound on them is known in the literature. As for Theorem 3, the above discussion on $\rho$ shows that it does not depend on the dimension directly. We will add a discussion on both $\rho$ and $\sigma^2_{p}$ in the final version of the paper, but please note that our use of these constants is in alignment with previous work, and an inspection of our proofs would show that we did not purposefully hide dimension dependence in these constants, but that rather they naturally arise from the problem.
>
> ---
>
> ***”Why are there no bounds on p=1, would there be a major complication in tackling this case?”***
>
> Yes, the case p=1 is technically more challenging, and we suspect that one needs new proof techniques to cover this case (as well as the case $p=\infty$). Our proof technique breaks since Lemma 4 does not hold for the case $p=1$. This is in part due to the vanishing of the second derivative of the absolute value function (where it exists), which prevents us from extracting a quadratic lower bound on the excess empirical risk. Please note that even the asymptotic behavior of the excess risk of ERM is quite difficult to obtain in the case $p=1$, and can widely vary depending on the degree of "non-realizability" of the problem; we refer the reviewer to (Knight, 1998) for a detailed account.
>
> ---
>
> ***”Bounds depend on Hessian forms of $\ell_{p}$ loss functions, which fit perfectly well to the quadratic function in the case $p=2$, but do not capture the loss as closely for other $p$, even though the bounds look similar.”***
>
> While the $\ell_{p}$ losses are indeed not quadratic for $p \neq 2$, in a small enough neighborhood of the optimum, the quadratic approximation of the risk is very accurate. Indeed, this is the basis of the asymptotic expansion in equation (1). As we have argued above, our bounds are optimal in that they asymptotically match the exact excess risk of ERM as $n \to \infty$, so that one cannot hope to obtain better bounds for large enough $n$. We agree however that for small $n$, better approximations of the $\ell_{p}$ loss could potentially lead to better bounds.
>
> ---
>
> ***”Discussion motivating the $\ell_{p}$ loss regression problem could be improved and seems vague wrt the properties of the model.”***
>
> We will include the reviewer’s suggestion in the final version of our paper as an additional motivation for $\ell_{p}$ norm regression. Please note however that the main perspective we take in this paper is learning theoretic, and our only concern is prediction: we are not postulating a statistical model, we make no assumptions on the distributional form of X and Y, and we are not motivating ERM as a maximum likelihood procedure.
>
> ---
>
> ***”What is a ‘negative moment’?”***
>
> For a random variable $X$, we call $E[X^{-k}]$ its $k$-th negative moment. (with the convention $1/0 = \infty$).
>
> ---
> We would be happy to clarify any concerns or answer any questions that may come up during the discussion period.

---

> > ### Comment · Reviewer_1SnJ · 2023-08-14
> >
> > I thank the authors for the response, which mostly clarified my questions.
> >
> > However, Question no. 2 is still open (the two examples of distributions where $\rho$ does not depend on the dimension do not resolve my concern). I assume $\delta$ is a constant, so it is out of the following discussion. Let me try to elaborate on what I think is wrong:
> >
> > - You claim "$O(d)$ samples are enough to exactly recover the target." in the abstract. To show this, you need to prove that for *any* distribution on (X,Y), O(d) samples suffice.
> > - In my Question no. 2, I mentioned that *there exists* a distribution that requires $\Omega(d\log d)$ samples to solve the task. This already contradicts the claim in the abstract (which should be modified).
> > - Assuming Theorem 3 is correct for *any* distribution (which I believe), the only way to make it compatible with the lower bound is when the remaining terms depending on $\rho$ are at least $\Omega(\log(d))$
> > - This implies that *there exists* a distribution where $\rho$ depends on the dimension (which I think needs to be discussed briefly in the paper).

---

> > > ### Author Response · Authors · 2023-08-14
> > >
> > > We thank the reviewer for following up with additional questions. Please find our response below.
> > >
> > > ---
> > >
> > > ***"Assuming Theorem 3 is correct for any distribution (which I believe), the only way to make it compatible with the lower bound is when the remaining terms depending on $\rho$ are at least $\Omega(\log{(d)})$. This implies that there exists a distribution where $\rho$ depends on the dimension (which I think needs to be discussed briefly in the paper)."***
> > >
> > > We agree with the reviewer. To be very precise, there exists a **sequence** of distributions, indexed by their dimension, for which $\rho$ grows with the dimension. We will briefly discuss this in the paper. However, please note that the spirit of our result is for a fixed but unknown distribution, and as our two examples above show, in such a general setting, $\rho$ does not depend on the dimension, which is why we didn't not mention this in our original submission.
> > >
> > > ---
> > >
> > > ***"You claim "$O(d)$ samples are enough to exactly recover the target." in the abstract. To show this, you need to prove that for any distribution on $(X,Y)$, $O(d)$ samples suffice. In my Question no. 2, I mentioned that there exists a distribution that requires $\Omega(d \log{d})$ samples to solve the task. This already contradicts the claim in the abstract (which should be modified)."***
> > >
> > > We agree with the reviewer that our statement needs to be modified. We used the $O$ notation to hide distribution specific constants but we did not specify this in the abstract, and as it stands, the current claim is imprecise. We will add the following sentence in the abstract to fix this:
> > >
> > > *"...,$O(d)$ samples are enough to exactly recover the target with high probability, where the $O$ notation hides dependence on a distribution specific constant as well as on the confidence level."*
> > >
> > > This revised statement does not contradict the reviewer’s example since the $\log{d}$ factor in their specific example should be recoverable from the distribution dependent constant $\rho$, which we are hiding in the $O$ notation.
> > >
> > > Please note that we initially tried to include the dependence on $\rho$ in the $O$ notation in the abstract. Unfortunately, this requires introducing the constant $\rho$, but more importantly, it requires listing the three cases of complexity of Theorem 3 to properly capture the dependence on $\rho$, which is quite cumbersome to do in the abstract. We settled on the current version, but we agree that we needed to clarify what we were hiding with the $O$ notation.
> > >
> > > We welcome any additional suggestion the reviewer might have in phrasing this statement precisely without encumbering the abstract.

---

> > > > ### Comment · Reviewer_1SnJ · 2023-08-15
> > > >
> > > > Thank you again for the very thoughtful and detailed responses. I agree with your suggestions on the wording.
> > > >
> > > > However, since you asked me for suggestions, I'd prefer "... , $O_\rho(d)$ samples are enough to exactly recover the target with high probability, where the $O_\rho$ notation hides dependence on a distribution specific *parameter* $\rho$ (as well as on the confidence level).", since I think *constant* is again suggesting plain $O(d)$, and the subscript notation is widely standard in the literature.
> > > >
> > > > Also, I will keep my initial rating, which was already on the positive end.

---

> > > > > ### Author Response · Authors · 2023-08-17
> > > > >
> > > > > We thank the reviewer again for the thoughtful discussion and valuable feedback. We will include their suggestion in the revised version of our paper.

---

### Decision · Program_Chairs · 2023-09-21

**Decision:**

Accept (poster)

**Comment:**

All reviewers agree to accept this paper. The paper is theoretically sound and sufficiently novel to grant acceptance at this conference.

For a camera-ready version, please take into the account the comments from all reviewers regarding typos and clarifications. In particular, please include:
- a discussion of optimality as understood in this paper versus minimax optimality
- intuition about the moment assumptions for the well-specified case and general case as discussed during the rebuttal